# Learning from Aggregate Observations

**Yivan Zhang**
The University of Tokyo / RIKEN
yivanzhang@ms.k.u-tokyo.ac.jp

**Nontawat Charoenphakdee**
The University of Tokyo / RIKEN
nontawat@ms.k.u-tokyo.ac.jp

**Zhenguo Wu**
The University of Tokyo
zhenguo@ms.k.u-tokyo.ac.jp

**Masashi Sugiyama**
RIKEN / The University of Tokyo
sugi@k.u-tokyo.ac.jp

## Abstract

We study the problem of *learning from aggregate observations* where supervision signals are given to *sets* of instances instead of individual instances, while the goal is still to predict labels of unseen individuals. A well-known example is multiple instance learning (MIL). In this paper, we extend MIL beyond binary classification to other problems such as multiclass classification and regression. We present a general probabilistic framework that accommodates a variety of aggregate observations, e.g., pairwise similarity/triplet comparison for classification and mean/difference/rank observation for regression. Simple maximum likelihood solutions can be applied to various differentiable models such as deep neural networks and gradient boosting machines. Moreover, we develop the concept of *consistency up to an equivalence relation* to characterize our estimator and show that it has nice convergence properties under mild assumptions. Experiments on three problem settings — *classification via triplet comparison* and *regression via mean/rank observation* indicate the effectiveness of the proposed method.

## 1 Introduction

Modern machine learning techniques usually require a large number of high-quality labels for *individual* instances [Jordan and Mitchell, 2015]. However, in the real world, it could be prohibited to obtain individual information while we can still obtain some forms of supervision for *sets* of instances. We give three following motivating use cases. The first scenario is when individual information cannot be released to the public due to *privacy concerns* [Horvitz and Mulligan, 2015]. One way to avoid releasing individual information is to only disclose some summary statistics of small groups to the public so that individual information can be protected. The second scenario is when individual annotations are *intrinsically unavailable* but group annotations are provided, which arises in problems such as drug activity prediction problem [Dietterich et al., 1997]. The third scenario is when the labeling cost for an individual instance is *expensive* [Zhou, 2017].

In classification, one of the most well-known examples of learning from aggregate observations is *multiple instance learning* (MIL) for binary classification [Zhou, 2004], where training instances are arranged in sets and the aggregate information is whether the positive instances exist in a set. MIL has found applications in many areas [Yang, 2005, Carbonneau et al., 2018], including aforementioned drug activity prediction [Dietterich et al., 1997]. *Learning from label proportions* (LLP) [Kück and de Freitas, 2005, Quadrianto et al., 2009, Yu et al., 2013, Patrini et al., 2014] is another well-studied example where the proportion of positive instances in each set is observed. However, most earlier studies as well as recent work in MIL and LLP only focus on binary classification and the type of aggregation is limited. Recently, *classification from comparison* has gained more attention. Bao et al. [2018] considered *learning from pairwise similarities*, where one can obtain a binary value

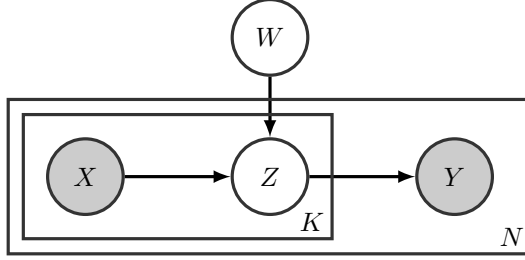

Figure 1: **Graphical representation of the data generating process.** $X$ represents the *feature vector* and $Z$ represents the unobservable *true target* for each $X$. An *aggregate observation* $Y$ can be observed from a set of $Z$, $Z_{1:K}$, via an *aggregate function* $T : \mathcal{Z}^K \mapsto \mathcal{Y}$, where $K \geq 2$ is the cardinality of the set. $Z$ follows a parametric distribution $p(Z|\theta)$ parameterized by $\theta = f(X; W)$, which is the output of a deterministic function $f$, parameterized by $W$. The goal is to estimate $W$ from $(X_{1:K}, Y)$-pairs to predict the true target $Z$ from a single feature vector $X$.

indicating whether two instances belong to the same category or not. Cui et al. [2020] considered *learning from triplets comparison*. Nevertheless, both works [Bao et al., 2018, Cui et al., 2020] are only applicable to binary classification. For multiclass classification from aggregate observations, we are only aware of a recent work by Hsu et al. [2019], where pairwise similarities are used to train a multiclass classifier based on maximum likelihood estimation, although its theoretical understanding is limited.

In regression, learning from aggregate observations has been assessed to only a limited extent. One example is *multiple instance regression* [Ray and Page, 2001, Amar et al., 2001], where it is assumed that there is only one instance in each set, called the "primary instance", that is responsible for the real-valued label. Another example is *learning from spatially aggregated data*, where only spatially summed or averaged values can be observed. Gaussian processes (GP) were proposed to handle this problem [Law et al., 2018, Tanaka et al., 2019a,b]. Although uncertainty information can be nicely obtained with a GP-based method, it may not be straightforward to scale it for high-dimensional data.

In general, the concept of drawing conclusions or making predictions about individual-level behavior based on aggregate-level data was also of great interest in other fields, such as ecological inference [Schuessler, 1999, King et al., 2004, Sheldon and Dietterich, 2011, Flaxman et al., 2015] and preference learning [Thurstone, 1927, Bradley and Terry, 1952, Luce, 1959, Plackett, 1975].

The goal of this paper is to provide a versatile framework that can be applied for many types of aggregate observations, both in multiclass classification and regression. We propose a simple yet effective solution to this class of problems based on maximum likelihood estimation, where Hsu et al. [2019]'s method is one of our special cases. Note that the maximum likelihood method has been explored for weak supervisions such as classification from noisy labels and coarse-grained labels [Patrini et al., 2017, Zhang et al., 2019], but only Hsu et al. [2019] considered aggregate observations.

Our contributions can be summarized as follows. Firstly, we expand the usage of maximum likelihood to aggregate observations, e.g., classification from triplet comparison and regression from mean/rank observation. Next, we provide a theoretical foundation of this problem by introducing the concept of *consistency up to an equivalence relation* to analyze the characteristics of our estimator. We demonstrate that this concept is highly useful by analyzing the behavior of our estimator and obtain insightful results, e.g., regression from mean observation is consistent but regression from rank observation is only consistent *up to an additive constant*. Finally, experimental results are also provided to validate the effectiveness of our method.

## 2 Problem setting

In this section, we clarify the notation and assumptions and introduce a probabilistic model for aggregate observations.

Table 1: **Examples of Learning from Aggregate Observations**

| Task | Learning from … | $K$ | Aggregate Observation $Y = T(Z_{1:K})$ |
|---|---|---|---|
| Classification $Z \in \{1, \dots, C\}$ | similarity/dissimilarity | $K = 2$ | if $Z_1$ and $Z_2$ are the same or not |
| | triplet comparison | $K = 3$ | if $d(Z_1, Z_2)$ is smaller than $d(Z_1, Z_3)$, where $d(\cdot, \cdot)$ is a similarity measure between classes |
| | multiple instance | $K \geq 2$ | if $Z_{1:K}$ contains positive instances ($C = 2$) |
| Regression $Z \in \mathbb{R}$ | mean/sum | $K \geq 2$ | the arithmetic mean or the sum of $Z_{1:K}$ |
| | difference/rank | $K = 2$ | the difference $Z_1 - Z_2$, or the relative order $Z_1 > Z_2$ |
| | min/max | $K \geq 2$ | the smallest/largest value in $Z_{1:K}$ |
| | uncoupled data | $K \geq 2$ | randomly permuted $Z_{1:K}$ |

## 2.1 Notation[1]

Let $X \in \mathcal{X}$ be the *feature vector*, and $Z \in \mathcal{Z}$ be the unobservable *true target* that we want to predict, where $\mathcal{X}$ and $\mathcal{Z}$ are their support spaces, respectively. The goal is to learn a discriminative model that predicts the true target $Z$ from the feature vector $X$. We do not have any restrictions on $\mathcal{Z}$ as long as we can model the conditional probability $p(Z|X)$. If $\mathcal{Z}$ is a finite set, e.g., $\{1, \dots, C\}$, the task is $C$-class classification; and if $\mathcal{Z}$ is $\mathbb{R}$, the task is regression. However, in the problem setting of learning from aggregate observations, we cannot observe the true target $Z$ directly. Instead, we assume that we can observe some information about *multiple instances*.

Concretely, the supervision signal $Y \in \mathcal{Y}$, called *aggregate observation*, for the set $Z_{1:K}$, can be obtained from $Z_{1:K}$ via an *aggregate function* $T : \mathcal{Z}^K \mapsto \mathcal{Y}$, i.e., $Y = T(Z_{1:K})$. We want to use observations of $(X_{1:K}, Y)$ to predict the true target $Z$ based on features $X$. Figure 1 illustrates the graphical representation of the data generating process.

## 2.2 Assumptions

Here, we summarize our assumptions used in learning from aggregate observations as follows.

**Assumption 1** (Aggregate observation assumption). $p(Y|X_{1:K}, Z_{1:K}) = p(Y|Z_{1:K})$.

This means that $Y$ and $X_{1:K}$ are conditionally independent given $Z_{1:K}$. This assumption is common in existing studies [Carbonneau et al., 2018, Hsu et al., 2019, Xu et al., 2019, Cui et al., 2020]. It can be implied in the data collection process, e.g., when we first collect $(X, Z)$-pairs but only disclose some summary statistics of $Z_{1:K}$ for learning due to privacy concerns. It also means that we expect the true target $Z$ to carry enough information about $X$, so that we do not need to extract more information from $X_{1:K}$ to predict $Y$.

Further, since we assumed that $Y$ can be determined by $Y = T(Z_{1:K})$, the conditional probability becomes $p(Y|Z_{1:K}) = \delta_{T(Z_{1:K})}(Y)$, where $\delta(\cdot)$ denotes the *Dirac delta function*.

**Assumption 2** (Independent observations assumption). $p(Z_{1:K}|X_{1:K}) = \prod_{i=1}^{K} p(Z_i|X_i)$.

This means that elements in $Z_{1:K}$ are mutually independent. It might be violated if we collect the data group by group, so that instances in a group tend to be similar and have high correlation [Carbonneau et al., 2018]. In such a case, our formulation and proposed method only serve as a practical approximation and our theoretical implications may not hold. Note that if the independence condition does not hold, aggregate observations could be more informative in some scenarios. For example, spatially aggregated data are aggregated within a specific region rather than randomly collected from many different regions [Law et al., 2018]. Thus, one may be able to utilize the dependency in $Z_{1:K}$ to obtain a better estimation of $p(Z_{1:K}|X_{1:K})$.

Combining these two assumptions, we can decompose the joint distribution $p(X_{1:K}, Z_{1:K}, Y)$ as

$$p(X_{1:K}, Z_{1:K}, Y) = p(Y|Z_{1:K}) \prod_{i=1}^{K} p(Z_i|X_i)p(X_i). \tag{1}$$

## 2.3 Aggregate function

The aggregate function $T : \mathcal{Z}^K \mapsto \mathcal{Y}$ characterizes different problems within our framework. It induces an *equivalence relation* $\sim$ over the set $\mathcal{Z}^K$ as $z_{1:K}^{(i)} \sim z_{1:K}^{(j)} \iff T(z_{1:K}^{(i)}) = T(z_{1:K}^{(j)})$. The coarseness of $\sim$ influences how hard the problem is. Since $\sim$ is usually much coarser than the equality equivalence relation on $\mathcal{Z}^K$, the model is not able to distinguish some observationally equivalent $Z_{1:K}$. However, we can analyze this equivalence relation $\sim$ and obtain the characteristics of the solution. In some problems, we are able to recover the conditional distribution $p(Z|X)$, or at least up to some transformation. We show this fact theoretically in Section 4 and experimentally in Section 6. In Table 1, we provide examples of problems that can be accommodated in our framework of learning from aggregate observations, which are categorized by underlying tasks and their aggregate functions.

## 3 Proposed method

In this section, we describe the proposed method for learning from aggregate observations.

The key idea is to estimate $p(Y|X_{1:K})$ based on $p(Z|X)$ using the aggregate function $T$. Based on the decomposition of the joint distribution in Equation (1), we can marginalize $p(Y, Z_{1:K}|X_{1:K}) = p(Y|Z_{1:K}) \prod_{i=1}^{K} p(Z_i|X_i)$ over $Z_{1:K}$ as

$$p(Y|X_{1:K}) = \int_{\mathcal{Z}^K} \delta_{T(z_{1:K})}(Y) \prod_{i=1}^{K} p(z_i|X_i) \, \mathrm{d}z_{1:K} = \mathop{\mathbb{E}}_{\substack{Z_i \sim p(Z_i|X_i) \\ i=1,\dots,K}} \left[ \delta_{T(Z_{1:K})}(Y) \right], \quad (2)$$

where $\mathbb{E}[\cdot]$ denotes the expectation.

To model the true target $Z$, we assume that $Z$ conditioned on $X$ follows a certain parametric distribution, parameterized by $\theta \in \Theta$, i.e., $p(Z_i|X_i) = p(Z_i|\theta = f(X_i; W))$. Here, $f : \mathcal{X} \to \Theta$ is a deterministic function parameterized by $W \in \mathcal{W}$, which maps the feature vector $X$ to the corresponding parameter $\theta$. Note that we do not restrict the family of $f$. It can be a deep neural network [LeCun et al., 2015] or a gradient boosting machine [Friedman, 2001], as long as the likelihood $p(Z|X)$ is differentiable w.r.t. the model parameter $W$.

Then, we use the likelihood to measure how likely our model with parameter $W$ can generate observed data. Our learning objective, the *expected log-likelihood*, is defined as

$$\ell(W) = \mathbb{E}[\log p(Y|X_{1:K}; W)]. \quad (3)$$

Approximating $\ell(W)$ based on an i.i.d. sample of $(X_{1:K}, Y)$-pair of size $N$, $\left\{ x_{1:K}^{(i)}, y^{(i)} \right\}_{i=1}^{N} \stackrel{\text{i.i.d.}}{\sim} p(X_{1:K}, Y)$, the *log-likelihood* is defined as

$$\ell_N(W) = \frac{1}{N} \sum_{i=1}^{N} \log p(y^{(i)}|x_{1:K}^{(i)}; W), \quad (4)$$

which converges to $\ell(W)$ almost surely as the sample size $N \to \infty$ [Van der Vaart, 2000]. Then, the *maximum likelihood estimator* (MLE) of $W$ given aggregate observations is defined as $\widehat{W}_N = \arg\max_W \ell_N(W)$.

Although MLE exists for all kinds of aggregate observations in theory, the likelihood can be hard to calculate because of the integral involved in Equation (2). In Section 5, we give three examples where we can obtain an analytical expression of the log-likelihood so that we do not need to resort to numerical integration or approximation methods to optimize the learning objective.

## 4 Consistency of learning from aggregate observations

In this section, we develop theoretical tools and show that although the estimator may not always converge to the true parameter, it can still capture some important information about the true parameter. In Section 5, we show that our theoretical analysis can elucidate the fact that our estimators can have different types of consistency depending on the given aggregate observations.

Assume that the parameter $W$ is in a metric space $(\mathcal{W}, d)$ with a metric $d$. Denote the true parameter by $W_0$. An estimator $\widehat{W}_N$ based on $N$ sample points is said to be *consistent* if $\widehat{W}_N \xrightarrow{\text{P}} W_0$ as $N \to \infty$, where $\xrightarrow{\text{P}}$ means the convergence in probability.

The MLE is consistent under mild conditions [Van der Vaart, 2000], but the *identifiability* of the model is always necessary. However, in learning from aggregate observations, it is more common that we can only *partially* identify the model. i.e., two parameter $W$ and $W'$ might be *observationally equivalent* and yield the same likelihood, which is defined as follows:

**Definition 3** (Equivalence). An *equivalence relation* $\sim$ on $\mathcal{W}$ induced by the likelihood is defined according to $W \sim W' \iff \ell(W) = \ell(W')$. The equivalence class of $W$ is denoted by $[W]$.

Then, *consistency up to an equivalence relation* is defined as follows:

**Definition 4** (Consistency up to $\sim$). An estimator $\widehat{W}_N$ is said to be *consistent up to an equivalence relation* $\sim$, if $d(\widehat{W}_N, [W_0]) \xrightarrow{\text{P}} 0$ as $N \to \infty$, where $d(W, [W_0]) = \inf_{W_0' \in [W_0]} d(W, W_0')$.

That is to say, an estimator converges in probability to a set of values that is at least observationally equivalent to the true parameter. Then, following Wald [1949] and Van der Vaart [2000], we can confirm that the MLE given aggregate observations has the desired behavior under mild assumptions:

**Proposition 5.** *The MLE $\widehat{W}_N$ based on Equation* (4) *is consistent up to $\sim$ if the following conditions hold:*

- *(A) The parameter space $\mathcal{W}$ is compact;*
- *(B) $\forall W \in \mathcal{W}$, the log-likelihood $\ell(W|x_{1:K}, y) = \log p(x_{1:K}, y; W)$ is upper-semicontinuous for almost all $(x_{1:K}, y)$;*
- *(C) $\forall$ sufficiently small ball $U \subset \mathcal{W}$, the function $\ell^U(W|x_{1:K}, y) = \sup_{W \in U} \ell(W|x_{1:K}, y)$ is measurable and satisfies $\mathbb{E}\big[\ell^U(W|X_{1:K}, Y)\big] < \infty$;*
- *(D) $\exists W_0' \in [W_0]$ s.t. $\ell_N(\widehat{W}_N) \geq \ell_N(W_0') - \delta_N$, where $\delta_N$ is a sequence of random variables such that $\delta_N \xrightarrow{\text{P}} 0$.*

Definitions 3, 4 and Proposition 5 provide a foundation to analyze the behavior of an MLE for learning from aggregate observations. The key difference between an analysis of the traditional MLE and ours is that due to limited supervision, the concept of *consistency* can be too strict. Although an MLE is not consistent for some aggregate observation (see Sections 5.2 and 5.4), we can show that an MLE is still useful by using the concept of *consistency up to an equivalence relation*.

# 5    Realizations of learning from aggregate observations

In this section, we illustrate three novel realizations of the proposed maximum likelihood approach to learning from aggregate observations where the integral in Equation (2) can be solved analytically. Furthermore, we demonstrate how to use the theoretical tools we developed in Section 4 to provide a better understanding of the behavior of the estimator given different kinds of aggregate observations.

Here, we will drop $\theta = f(X; W)$ from expressions to keep the notation uncluttered. Any parameter of the distribution of $Z$ is either determined from $X$ in this way or just kept fixed. We defer the proofs of propositions in this section to Appendix A.

## 5.1    Classification via pairwise similarity

In classification, the true target is categorical: $Z \in \{1, \ldots, C\}$, where $C$ is the number of classes. We model it using a categorical distribution $Z \sim \text{Categorical}(p_{1:C})$, with probability parameters $p_i > 0$, $\sum_{i=1}^C p_i = 1$. Because $|\mathcal{Z}^K| = C^K$ is always finite, the integration in Equation (2) becomes a summation and the log-likelihood is always analytically calculable.

**Pairwise similarity** is our first example of learning from aggregate observations. In this problem, we only know if a pair of instances belongs to the same class or not. Here $K = 2$. Concretely,

$$Y = T_{\text{sim}}(Z_1, Z_2) = [Z_1 = Z_2], \quad p(Y = 1) = \sum_{i=1}^C p(Z_1 = i)p(Z_2 = i), \tag{5}$$

where $[\cdot]$ denotes the Iverson bracket.[2] This problem is also considered to be *semi-supervised clustering* in the literature [Xing et al., 2003, Basu et al., 2004, Bilenko et al., 2004, Davis et al., 2007]. Recently, Hsu et al. [2019] studied this problem thoroughly for several image classification tasks from the perspective of classification, whose method is also based on the maximum likelihood and thus a special case within our framework.

## 5.2 Classification via triplet comparison

**Triplet comparison** is of the form "$A$ (*anchor*) is more similar to $P$ (*positive*) than $N$ (*negative*)." Assuming that $d : \mathcal{Z} \times \mathcal{Z} \mapsto \mathbb{R}$ is a similarity measure between classes, we can instantiate the aggregation function $T$ and Equation (2) for $K = 3$:

$$Y = T_{\text{tri}}(Z_1, Z_2, Z_3) = [d(Z_1, Z_2) < d(Z_1, Z_3)],$$

$$p(Y = 1) = \sum_{\substack{d(i,j) < d(i,k) \\ i,j,k \in \{1,\ldots,C\}}} p(Z_1 = i)p(Z_2 = j)p(Z_3 = k). \tag{6}$$

Based on Equation (4), the loss function for triplet comparison is in the form of binary cross entropy:

$$
\begin{aligned}
\ell_N^{\text{tri}}(W) = &-\frac{1}{N} \sum_{i=1}^N [y^{(i)} = 1] \log p(y^{(i)} = 1 | x_{1:K}^{(i)}; W) \\
&-\frac{1}{N} \sum_{i=1}^N [y^{(i)} = 0] \log p(y^{(i)} = 0 | x_{1:K}^{(i)}; W).
\end{aligned}
\tag{7}
$$

Although triplet comparison data has been used for *metric learning* and *representation learning* [Sohn, 2016, Schroff et al., 2015, Mojsilovic and Ukkonen, 2019, Schultz and Joachims, 2004, Kumar and Kummamuru, 2008, Tamuz et al., 2011, Baghshah and Shouraki, 2010], studies on classification based on solely triplet comparison remain limited. Cui et al. [2020] recently showed that it is possible to learn a binary classifier from triplet comparison directly. But so far no result has been given for the multiclass setting, to the best of our knowledge. Existing work either requires labeled data [Perrot and Von Luxburg, 2019] or needs to actively inquire comparison labels from an oracle [Haghiri et al., 2018].

Since the parameters of the model given only the pairwise similarity or the triplet comparison are not identifiable, the corresponding MLE cannot be consistent. But that does not mean the learned model does not capture any information. Based on our framework, we prove the following proposition, which suggests that a multiclass classifier is still learnable at most up to a permutation of classes.

**Proposition 6** (Classification via triplet comparison is at most consistent up to a permutation)**.** *Let* $f(X; W)$ *be a $C$-dimensional vector of probability parameters $p_{1:C}$ for classification, then*

$$\{W \in \mathcal{W} : \exists \text{ permutation matrix } P, \text{ s.t. } Pf(X; W) = f(X; W_0) \text{ a.e.}\} \subseteq [W_0]. \tag{8}$$

Proposition 6 states that the estimator can be consistent up to a permutation, but *not always*. Please refer to Appendix A for more discussion. The same result holds for the pairwise similarity case, which was investigated empirically by Hsu et al. [2019]. In Section 6.1, we show that the proposed method works well empirically when the true labels are not too ambiguous.

## 5.3 Regression via mean observation

In regression problems, the true target is a real value $Z \in \mathbb{R}$. The *mean squared error* (MSE) is the canonical choice for the loss function, which can be derived from the maximum likelihood under an *additive homoscedastic Gaussian noise* model [Bishop, 2006]. Thus, we also use a Gaussian distribution for the true target, $Z \sim \mathcal{N}(\mu, \sigma^2)$ with mean $\mu \in \mathbb{R}$ and standard deviation $\sigma \in (0, \infty)$.

Gaussian distributions have desired properties, including *stability* and *decomposability*. Therefore Gaussian distribution is closed under linear combination given independence (See Appendix B). We

can model several aggregate observations, e.g., the mean observation and the rank observation, and obtain an analytical expression of the log-likelihood, as discussed below.

**Mean observation** is the arithmetic mean of the set of true targets $Z_{1:K}$. Under the Gaussian distribution assumption, the mean $Y$ is also a Gaussian random variable:

$$Y = T_{\text{mean}}(Z_{1:K}) = \frac{1}{K} \sum_{i=1}^{K} Z_i, \quad Y \sim \mathcal{N}\left( \frac{1}{K} \sum_{i=1}^{K} \mu_i, \frac{1}{K^2} \sum_{i=1}^{K} \sigma_i^2 \right). \tag{9}$$

Assuming homoscedastic noise, $Z_i = f(X_i; W) + \varepsilon_i$, for $i = 1, \dots, K$, where $\varepsilon_i \overset{\text{i.i.d.}}{\sim} \mathcal{N}(0, \sigma^2)$, the loss function realized from Equation (4) becomes

$$\ell_N^{\text{mean}}(W) = \frac{1}{N} \sum_{i=1}^{N} \left( y^{(i)} - \frac{1}{K} \sum_{j=1}^{K} f(x_j^{(i)}; W) \right)^2. \tag{10}$$

Theoretically, we can obtain the following proposition, which states that our estimator is consistent:

**Proposition 7** (Regression via mean observation is consistent). *The MLE $\widehat{W}_N$ based on Equation (4) for mean observations obtained by Equation (9) is consistent, because*

$$[W_0] = \{W \in \mathcal{W} : f(X; W) = f(X; W_0) \text{ a.e.}\}. \tag{11}$$

In Section 6.2, we validate Proposition 7 and demonstrate that it is feasible to learn solely from mean observations using our method and achieve MSE that is comparable to learning from direct observations. In Appendix C, we provide possible use of the Cauchy distribution for robust regression and the Poisson distribution for count data in addition to the Gaussian distribution.

### 5.4 Regression via rank observation

**Rank observation**, or called *pairwise comparison* in the context of regression, indicates the relative order between two real values. Since the difference of two Gaussian random variables is still Gaussian distributed, the likelihood of the rank can be derived from the cumulative distribution function of a Gaussian distribution:

$$Y = T_{\text{rank}}(Z_1, Z_2) = [Z_1 > Z_2], \quad p(Z_1 > Z_2) = \frac{1}{2}\left[ 1 + \text{erf}\left( \frac{\mu_1 - \mu_2}{\sqrt{2(\sigma_1^2 + \sigma_2^2)}} \right) \right], \tag{12}$$

where $\text{erf}(\cdot)$ is the error function. Note that even though $\text{erf}(\cdot)$ is a special function, its value can be approximated and its gradient is analytically calculable.[3] Thus the gradient-based optimization is still applicable.

Without loss of generality, we assume $z_1^{(i)} > z_2^{(i)}$ in the dataset. For a fixed variance $\sigma^2$, the loss function derived from Equations (4) and (12) is

$$\ell_N^{\text{rank}}(W) = -\frac{1}{N} \sum_{i=1}^{N} \log \frac{1}{2}\left[ 1 + \text{erf}\left( \frac{f(x_1^{(i)}; W) - f(x_2^{(i)}; W)}{2\sigma} \right) \right]. \tag{13}$$

The rank observation has been studied recently in Xu et al. [2019] to solve an uncoupled regression problem, where additional information is required. Here, an important question is how good the estimator can be if we only have pairwise comparison. Intuitively, it should not be consistent. However, surprisingly, it can be theoretically showed that it is still possible to learn solely from rank observations, and the prediction and the true target differ only by a constant under mild conditions.

**Proposition 8** (Regression via rank observation is consistent up to an additive constant). *Assuming homoscedastic noise, $Z_i = f(X_i; W) + \varepsilon_i$ for $i = 1, \dots, K$, where $\varepsilon_i \overset{\text{i.i.d.}}{\sim} \mathcal{N}(0, \sigma^2)$, the MLE $\widehat{W}_N$ based on Equation (4) for rank observations obtained by Equation (12) is consistent up to an additive constant, because*

$$[W_0] = \{W \in \mathcal{W} : \exists C \in \mathbb{R}, f(X; W) - f(X; W_0) = C \text{ a.e.}\}. \tag{14}$$

Table 2: **Classification via pairwise similarity and triplet comparison.** Means and standard deviations of accuracy (after the optimal permutation) in percentage for 10 trials are reported.

| Dataset | Unsupervised | Pairwise Similarity | | | Triplet Comparison | | | Supervised |
|---|---|---|---|---|---|---|---|---|
| | | Siamese | Contrastive | Ours[*] | Tuplet | Triplet | Ours | |
| MNIST | 52.30 | **85.82** | 98.45 | **98.84** | 18.42 | 22.77 | **94.94** | 99.04 |
| | (1.15) | (**24.86**) | (0.11) | (**0.10**) | (1.08) | (9.38) | (**3.68**) | (0.08) |
| FMNIST | 50.94 | 62.86 | 88.49 | **90.59** | 21.98 | 27.27 | **81.49** | 91.97 |
| | (3.28) | (17.97) | (0.28) | (**0.26**) | (0.72) | (12.82) | (**0.94**) | (0.24) |
| KMNIST | 40.22 | 61.30 | 89.65 | **93.45** | 16.00 | 20.39 | **81.94** | 94.47 |
| | (0.01) | (17.41) | (0.19) | (**0.32**) | (0.27) | (2.03) | (**4.59**) | (0.21) |

[*] See also Hsu et al. [2019].

As a result, we can guarantee that if we know the mean or can obtain a few direct observations, precise prediction becomes possible. In Section 6.2 we validate this finding. In Appendix D, we provide possible use of the Gumbel distribution, Cauchy distribution, and Exponential distribution for regression via rank observation.

# 6 Experiments

In this section, we present the empirical results of the proposed method. All experimental details can be found in Appendix E. More extensive experiments on 20 regression datasets and 30 classification datasets are provided in Appendix F. For each type of aggregate observations, outperforming methods are highlighted in boldface using one-sided t-test with a significance level of 5%.

## 6.1 Classification via triplet comparison

We demonstrate that a multiclass classifier is learnable using only triplet comparison data introduced in Section 5.2. We evaluate our method on three image classification datasets, namely MNIST,[4] Fashion-MNIST (FMNIST),[5] and Kuzushiji-MNIST (KMNIST),[6] which consist of $28 \times 28$ grayscale images in 10 classes.

As for the similarity measure $d$, we followed Cui et al. [2020] and simply used $d = [Z_i \neq Z_j]$ as the similarity measure between classes. We also compared learning from pairwise similarity data, which was studied in Hsu et al. [2019]. Both pairwise similarity and triplet comparison observations were generated according to our assumptions in Section 2.2. Since both learning from pairwise similarities and triplet comparisons are only consistent *up to a permutation* at best, we followed Hsu et al. [2019] and evaluated the performance by modified accuracy that allows any permutation of classes. The optimal permutation is obtained by solving a linear sum assignment problem using a small amount of individually labeled data [Kuhn, 1955].

**Baseline.** We used K-means as the unsupervised clustering baseline and the fully supervised learning method as a reference. We also compared representation learning and metric learning methods such as the Siamese network [Koch et al., 2015] and contrastive loss [Hadsell et al., 2006] for pairwise similarity, the (2+1)-tuplet loss [Sohn, 2016] and the triplet loss [Schroff et al., 2015] for triplet comparison. Since the output of such methods is a vector representation, we performed K-means to obtain a prediction so that the results can be directly compared.

**Results.** Table 2 shows that our method outperforms representation learning methods. It demonstrates that if the goal is classification, directly optimizing a classification-oriented loss is better than combining representation learning and clustering. Representation learning from triplet comparison also suffers from a lack of data because a large amount of triplet comparison data could be in the form of "$Z_1$ is equally similar/dissimilar to $Z_2$ and $Z_3$" if either $Z_2$ and $Z_3$ belong to the same class or $Z_{1:3}$ belong to three different classes. Such data cannot be used for representation learning but still can provide some information for the classification task.

Table 3: **Regression via mean observation and rank observation on UCI benchmark datasets.** Means and standard deviations of error variance (rank observations) or MSE (otherwise) for 10 trials are reported. We compare linear regression (LR) and gradient boosting machines (GBM) as the regression function.

| Dataset | Mean Observation | | | | Rank Observation | | | | Supervised | |
|---|---|---|---|---|---|---|---|---|---|---|
| | Baseline | | Ours | | RankNet, Gumbel | | Ours, Gaussian | | | |
| | LR | GBM | LR | GBM | LR | GBM | LR | GBM | LR | GBM |
| abalone | 7.91 (0.4) | 7.89 (0.5) | 5.27 (0.4) | **4.80** (**0.3**) | 5.81 (0.4) | 10.66 (0.7) | **5.30** (**0.3**) | **5.04** (**0.5**) | 5.00 (0.3) | 4.74 (0.4) |
| airfoil | 38.57 (2.0) | 28.65 (2.5) | 23.59 (1.8) | **4.63** (**0.9**) | 37.15 (1.8) | 47.46 (3.7) | 27.95 (1.1) | **6.18** (**1.0**) | 22.59 (1.9) | 3.84 (0.5) |
| auto-mpg | 41.59 (5.7) | 36.31 (1.9) | 14.61 (3.2) | **9.53** (**2.4**) | 27.26 (4.0) | 65.39 (7.4) | 17.34 (2.0) | **9.97** (**2.0**) | 11.73 (2.3) | 7.91 (1.6) |
| concrete | 198.51 (12.8) | 172.35 (15.2) | 115.06 (10.1) | **31.84** (**3.0**) | 244.06 (17.1) | 268.86 (26.5) | 233.93 (20.0) | **38.11** (**5.4**) | 111.92 (6.4) | 24.80 (5.7) |
| housing | 67.40 (20.8) | 52.23 (6.0) | 27.54 (6.8) | **14.85** (**3.0**) | 52.51 (10.8) | 93.07 (8.1) | 44.40 (13.4) | **23.49** (**6.9**) | 29.66 (6.1) | 13.12 (3.7) |
| power-plant | 172.64 (7.1) | 170.10 (3.4) | 20.73 (0.8) | **12.82** (**0.6**) | 163.64 (4.8) | 294.07 (4.9) | 44.82 (6.1) | **26.06** (**2.5**) | 21.17 (1.0) | 11.84 (0.9) |

## 6.2 Regression via mean/rank observation

We compare the performance in regression via direct/mean/rank observations introduced in Sections 5.3 and 5.4. We present results on benchmark datasets to show the effectiveness of our method.

**Baseline.** For the mean observation, we used a method treating the mean as the true label for each instance as the baseline. For the rank observation, we used RankNet [Burges et al., 2005] for regression to compare different distribution hypotheses.

**Real-world dataset.** We conducted experiments on 6 UCI benchmark datasets[7] and compared linear models and gradient boosting machines as the regression function. Mean observations of four instances and rank observations are generated according to our assumptions in Section 2.2. Since learning from rank observations is only consistent up to an additive constant, we measured the performance by modified MSE that allows any constant shift. This metric coincides with the variance of the error. If the estimator is unbiased, it is equal to MSE. Concretely,

$$\min_C \frac{1}{N} \sum_{i=1}^N \Big( Z_i - (\widehat{Z}_i + C) \Big)^2 = \mathrm{Var}[Z - \widehat{Z}]. \tag{15}$$

**Results.** In Table 3, we report error variance for learning from rank observations, otherwise MSE. It shows that learning from mean observations achieved MSE that is comparable to learning from direct observations, while the error variance of learn from rank observations is slightly higher. Further, our method consistently outperforms the baseline for the mean observation and RankNet for regression via rank observation. The performance ranking of learning from direct/mean/rank observations is roughly maintained regardless of the complexity of the model, while the gradient boosting machine usually performs better than the linear model.

## 7 Conclusions

We presented a framework for *learning from aggregate observations*, where only supervision signals given to sets of instances are available. We proposed a simple yet effective method based on the *maximum likelihood* principle, which can be simply implemented for various differentiable models, including deep neural networks and gradient boosting machines. We also theoretically analyzed the characteristic of our proposed method based on the concept of consistency up to an equivalent relation. Experiments on classification via pairwise similarity/triplet comparison and regression via mean/rank observation suggested the feasibility and the usefulness of our method.

## 8 Broader impact

In this work we proposed a general method that learns from supervision signals given to sets of instances. Such studies could provide new tools for privacy preserving machine learning because individual labels are not needed. This leads to a new way to anonymize data and alleviate potential threats to privacy-sensitive information, in addition to well-known differential privacy techniques [Dwork et al., 2014], which inject noise into the data to guarantee the anonymity of individuals.

However, such studies may have some negative consequences because depending on the type of aggregate observations, in the worst case, it may be possible to uncover individual information to some extent from aggregated statistics, even if individual labels are not available in the training data. Nevertheless, a person can deny the uncovered label because of a lack of evidence. So learning from aggregate observations is still arguably safer than the fully-supervised counterparts in terms of privacy preservation.

Our framework also opens possibilities of using machine learning technology in new problem domains where true labels cannot be straightforwardly obtained, but information such as pairwise/triplet comparisons or coarse-grained data are available or possible to collect. Finally, we believe that theoretical understanding could provide a better foundation towards solving learning from aggregate observations more effectively in the future.

## Acknowledgement

We thank Zhenghang Cui, Takuya Shimada, Liyuan Xu, and Zijian Xu for helpful discussion. We also would like to thank the Supercomputing Division, Information Technology Center, The University of Tokyo, for providing the Reedbush supercomputer system. NC was supported by MEXT scholarship, JST AIP Challenge, and Google PhD Fellowship program. MS was supported by JST CREST Grant Number JPMJCR18A2.

## Footnotes

[1]In this work, we denote uppercase letters $X, Y, Z$ as random variables, lowercase letters $x, y, z$ as instances of random variables, and calligraphic letters $\mathcal{X}, \mathcal{Y}, \mathcal{Z}$ as their support spaces. The subscript such as $Z_{1:K}$ is an abbreviation for the set $\{Z_1, Z_2, \dots, Z_K\}$. The superscript such as $y^{(i)}$ denotes the $i$-th sample point in a dataset. With abuse of notation, $p(\cdot)$ denotes a distribution and also its probability mass/density function.

[2]Iverson bracket $[\cdot]$: for any logic expression $P$, $[P] = 1$ if $P$ is true, otherwise $0$.

[3]$\text{erf}(x) = \frac{1}{\sqrt{\pi}} \int_{-x}^{x} e^{-t^2} \, dt$, and $\frac{d}{dx} \text{erf}(x) = \frac{2}{\sqrt{\pi}} e^{-x^2}$ [Andrews, 1998, p.110].

[4]MNIST [LeCun et al., 1998] `http://yann.lecun.com/exdb/mnist/`

[5]Fashion-MNIST [Xiao et al., 2017] `https://github.com/zalandoresearch/fashion-mnist`

[6]Kuzushiji-MNIST [Clanuwat et al., 2018] `http://codh.rois.ac.jp/kmnist/`

[7]UCI Machine Learning Repository [Dua and Graff, 2017] `https://archive.ics.uci.edu`

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
