[Supplementary Material]

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

# Appendix

## A  Proofs

In this section, we provide missing proofs of Proposition 6, Proposition 7, and Proposition 8.

### A.1  Proof of Proposition 6

*Proof.* Denote $X' = (X_1, X_2, X_3)$ where $X_1, X_2, X_3$ are i.i.d. observations. Let $p_i(X', W) = p(Y = i | X_1, X_2, X_3; W)$, $i = 0, 1$.

Following simple algebra,

$$\ell(W) = \mathbb{E}[\mathbb{E}[\log p(Y|X', W)|X']] = \mathbb{E}\left[\sum_{i=0}^{1} p_i(X', W_0) \log p_i(X', W)\right]. \tag{16}$$

So, the difference of $\ell(W)$ and $\ell(W_0)$ satisfies

$$\begin{aligned}
\ell(W) - \ell(W_0) &= \mathbb{E}\left[\sum_{i=0}^{1} p_i(X', W_0) \log \frac{p_i(X', W)}{p_i(X', W_0)}\right] \\
&\leq \mathbb{E}\left[\sum_{i=0}^{1} p_i(X', W_0)\left(\frac{p_i(X', W)}{p_i(X', W_0)} - 1\right)\right] \\
&= \mathbb{E}\left[\sum_{i=0}^{1} p_i(X', W) - \sum_{i=0}^{1} p_i(X', W_0)\right] \\
&= 0.
\end{aligned} \tag{17}$$

The inequality holds because $\log x \leq x - 1$ for $x > 0$, where "$=$" holds if and only if $x = 1$. Hence $\ell(W) = \ell(W_0)$, which implies that $p_i(X', W) = p_i(X', W_0)$, a.e. for $i = 0, 1$. This is equivalent to $p_1(X', W) = p_1(X', W_0)$.

Let $\sigma$ be a permutation function of $\{1, 2, \ldots, C\}$. That is, $\{\sigma(1), \sigma(2), \ldots, \sigma(C)\} = \{1, 2, \ldots, K\}$. If $p(Z = i | X; W_0) = p(Z = \sigma(i) | X; W)$, $\forall 1 \leq i \leq K$. With the assumption $d(\sigma(i), \sigma(j)) = d(i, j)$, we have

$$\begin{aligned}
p_1(X', W) &= \sum_{\substack{d(i,j)<d(i,k) \\ i,j,k \in \{1,\ldots,C\}}} p(Z_1 = i|X_1; W)p(Z_2 = j|X_2; W)p(Z_3 = k|X_3; W) \\
&= \sum_{\substack{d(\sigma(i),\sigma(j))<d(\sigma(i),\sigma(k)) \\ \sigma(i),\sigma(j),\sigma(k) \in \{1,\ldots,C\}}} p(Z_1 = \sigma(i)|X_1; W)p(Z_2 = \sigma(j)|X_2; W)p(Z_3 = \sigma(k)|X_3; W) \\
&= \sum_{\substack{d(\sigma(i),\sigma(j))<d(\sigma(i),\sigma(k)) \\ \sigma(i),\sigma(j),\sigma(k) \in \{1,\ldots,C\}}} p(Z_1 = i|X_1; W)p(Z_2 = j|X_2; W)p(Z_3 = k|X_3; W) \\
&= \sum_{\substack{d(i,j)<d(i,k) \\ \sigma(i),\sigma(j),\sigma(k) \in \{1,\ldots,C\}}} p(Z_1 = i|X_1; W)p(Z_2 = j|X_2; W)p(Z_3 = k|X_3; W) \\
&= \sum_{\substack{d(i,j)<d(i,k) \\ i,j,k \in \{1,\ldots,C\}}} p(Z_1 = i|X_1; W)p(Z_2 = j|X_2; W)p(Z_3 = k|X_3; W) = p_1(X', W_0).
\end{aligned}$$

On the other hand, if we consider a special case where $C = 3$, distance function $d(i, j) = [i \neq j]$, and $(p(Z = 1|X; W), p(Z = 2|X; W), p(Z = 3|X; W)) = \left(\frac{1}{3} - W, \frac{1}{3}, \frac{1}{3} + W\right)$, then the following

equation always holds:

$$
\begin{aligned}
p_1(X', W) &= \sum_{\substack{i=j, i\neq k \\ i,j,k \in \{1,2,3\}}} p(Z_1 = i|X_1; W)p(Z_2 = j|X_2; W)p(Z_3 = k|X_3; W) \\
&= \sum_{i=1}^{3} \sum_{k\neq i} p(Z_1 = i|X_1; W)p(Z_2 = i|X_2; W)p(Z_3 = k|X_3; W) \\
&= \sum_{i=1}^{3} p(Z_1 = i|X_1; W)p(Z_2 = i|X_2; W)(1 - p(Z_3 = i|X_3; W)) \\
&= \left(\frac{1}{3} - W\right)^2 \left(\frac{2}{3} + W\right) + \left(\frac{1}{3}\right)^2 \left(\frac{2}{3}\right) + \left(\frac{1}{3} + W\right)^2 \left(\frac{2}{3} - W\right) \\
&= \frac{2}{9}.
\end{aligned}
$$

In this example, $\forall W_0 \in [0, \frac{1}{3}]$, equivalence up to a permutation requires $W = W_0$ or $W = \frac{2}{3} - W_0$ but $[W_0] = [0, \frac{1}{3}]$. Hence, this counterexample shows that little information can be learned from triplet comparison if the distance function or the distribution of $Z$ is not modeled properly.

$\square$

Further, we can also prove that classification via pairwise similarity is at most consistent up to a permutation, which is empirically shown in Hsu et al. [2019]. More concretely, we can show that it is consistent up to rotations and reflections in the parameter space.

*Proof.* Let $p(Z_i|X_i) \in \Delta^{C-1}$, $i = 1, 2$, be the vector of probability parameters of a categorical distribution, where $\Delta^{C-1}$ denotes $C - 1$-dimensional simplex. Then the probability of similarity $Y = [Z_1 = Z_2]$, i.e., $X_1$ and $X_2$ belong to the same class, can be written as

$$p(Y = 1|X_1, X_2) = p(Z_1|X_1)^\mathsf{T} p(Z_2|X_2). \tag{18}$$

From $p(Y = 1|X_1, X_2)$ we cannot identify $p(Z_1|X_1)$ and $p(Z_2|X_2)$, because for any orthogonal matrix $Q$, representing rotations and reflections in the parameter space,

$$(Qp(Z_1|X_1))^\mathsf{T} (Qp(Z_2|X_2)) = p(Z_1|X_1)^\mathsf{T} p(Z_2|X_2) \tag{19}$$

holds for any $p(Z_1|X_1)$ and $p(Z_2|X_2)$. The choice of $Q$ is restricted because $Qp(Z_1|X_1)$ and $Qp(Z_2|Z_2)$ should be on the simplex $\Delta^{C-1}$ too. Any permutation matrix $P$ is an orthogonal matrix and always satisfies this condition. So classification via pairwise similarity is at most consistent up to a permutation.

$\square$

## A.2 Proof of Proposition 7

*Proof.* Given $X_{1:K}$, $Y = \frac{1}{K}\sum_{i=1}^{K} Z_i \sim \mathcal{N}\left(\frac{1}{K}\sum_{i=1}^{K} f(X_i; W_0), \frac{\sigma^2}{K}\right)$.

Denote $\mu(X_{1:K}, W) = \frac{1}{K}\sum_{i=1}^{K} f(X_i; W)$.

We have

$$\ell(W) = \mathbb{E}[\mathbb{E}[\log p(Y|X_{1:K}; W)|X_{1:K}]] = -\frac{K}{2\sigma^2}\mathbb{E}\big[\mathbb{E}\big[(Y - \mu(X_{1:K}, W))^2|X_{1:K}\big]\big] + C, \tag{20}$$

where $C$ is a constant.

Noting that

$$
\begin{aligned}
&\mathbb{E}\big[(Y - \mu(X_{1:K}, W)^2|X_{1:K}\big] \\
=\,& \mathbb{E}\big[(Y - \mu(X_{1:K}, W_0) + \mu(X_{1:K}, W_0) - \mu(X_{1:K}, W))^2|X_{1:K}\big] \\
=\,& \mathbb{E}\big[(Y - \mu(X_{1:K}, W_0))^2|X_{1:K}\big] + \mathbb{E}\big[(\mu(X_{1:K}, W_0) - \mu(X_{1:K}, W))^2|X_{1:K}\big] \\
&+ 2\,\mathbb{E}[(Y - \mu(X_{1:K}, W_0))(\mu(X_{1:K}, W_0) - \mu(X_{1:K}, W))|X_{1:K}] \\
=\,& \mathbb{E}\big[(Y - \mu(X_{1:K}, W_0))^2|X_{1:K}\big] + (\mu(X_{1:K}, W_0) - \mu(X_{1:K}, W))^2 \\
&+ 2(\mu(X_{1:K}, W_0) - \mu(X_{1:K}, W))\,\mathbb{E}[Y - \mu(X_{1:K}, W_0)|X_{1:K}] \\
=\,& \mathbb{E}\big[(Y - \mu(X_{1:K}, W_0))^2|X_{1:K}\big] + (\mu(X_{1:K}, W_0) - \mu(X_{1:K}, W))^2,
\end{aligned}
\tag{21}
$$

$\ell(W) = \ell(W_0)$ implies that

$$
\mathbb{E}\left[(\sum_{i=1}^{K} f(X_i; W) - \sum_{i=1}^{K} f(X_i; W_0))^2\right] = 0.
\tag{22}
$$

Denote $f(X; W) - f(X; W_0)$ as $\Delta(X, W)$. Equation (22) can be expanded as follows

$$
\begin{aligned}
0 =\,& \mathbb{E}\left[(\sum_{i=1}^{K} \Delta(X_i, W))^2\right] \\
=\,& \sum_{i=1}^{K} \mathbb{E}\big[\Delta^2(X_i, W)\big] + 2 \sum_{1 \le i < j \le K} \mathbb{E}[\Delta(X_i, W)\Delta(X_j, W)] \\
=\,& \sum_{i=1}^{K} \mathbb{E}\big[\Delta^2(X_i, W)\big] + 2 \sum_{1 \le i < j \le K} \mathbb{E}[\Delta(X_i, W)]\,\mathbb{E}[\Delta(X_j, W)] \\
=\,& \sum_{i=1}^{K} \mathbb{E}\big[\Delta^2(X_i, W)\big] + 2 \sum_{1 \le i < j \le K} (\mathbb{E}[\Delta(X_i, W)])^2.
\end{aligned}
\tag{23}
$$

Thus, $\mathbb{E}\big[\Delta^2(X_i, W)\big] = \mathbb{E}[\Delta(X_i, W)] = 0$. We have $\Delta(X_i, W) = 0$, a.e., therefore $f(X, W) = f(X, W_0)$, a.e.

$\square$

### A.3 Proof of Proposition 8

*Proof.* Denote $X' = (X_1, X_2)$ and denote $p(Y = i|X_1, X_2; W)$ as $p_i(X', W), i = 0, 1$.

Following simple algebra,

$$
\ell(W) = \mathbb{E}[\mathbb{E}[\log p(Y|X', W)|X']] = \mathbb{E}\left[\sum_{i=0}^{1} p_i(X', W_0) \log p_i(X', W)\right].
\tag{24}
$$

So, the difference of $\ell(W)$ and $\ell(W_0)$ satisfies

$$
\begin{aligned}
\ell(W) - \ell(W_0) =\,& \mathbb{E}\left[\sum_{i=0}^{1} p_i(X', W_0) \log \frac{p_i(X', W)}{p_i(X', W_0)}\right] \\
\le\,& \mathbb{E}\left[\sum_{i=0}^{1} p_i(X', W_0)\left(\frac{p_i(X', W)}{p_i(X', W_0)} - 1\right)\right] \\
=\,& \mathbb{E}\left[\sum_{i=0}^{1} p_i(X', W) - \sum_{i=0}^{1} p_i(X', W_0)\right] \\
=\,& 0.
\end{aligned}
\tag{25}
$$

The inequality holds because $\log x \leq x - 1$ for $x > 0$, where "=" holds if and only if $x = 1$. Hence $\ell(W) = \ell(W_0)$, which implies that $p_i(X', W) = p_i(X', W_0)$, a.e. for $i = 0, 1$. Together with Equation (12), we have

$$\text{erf}\left(\frac{1}{2\sigma}(f(X_1, W) - f(X_2, W))\right) = \text{erf}\left(\frac{1}{2\sigma}(f(X_1, W_0) - f(X_2, W_0))\right). \quad (26)$$

Hence, $f(X_1, W) - f(X_2, W) = f(X_1, W_0) - f(X_2, W_0)$, a.e.

$\square$

# B  Preliminary: mean squared error and Gaussian distribution

In this section, we provide preliminaries related to Section 5.3 and Section 5.4.

## B.1  Mean squared error

It is well known that the *mean squared error* (MSE) loss function can be derived from the maximum likelihood estimation under an *additive homoscedastic Gaussian noise* model [see also Bishop, 2006, p. 140]. Concretely, let the target $Z$ be given by a deterministic function $f(X; W)$ with additive Gaussian noise $\varepsilon$ with zero mean and fixed variance:

$$Z = f(X; W) + \varepsilon, \text{ where } \varepsilon \sim \mathcal{N}(0, \sigma^2). \quad (27)$$

Or equivalently,

$$Z \sim \mathcal{N}(\mu, \sigma^2), \text{ where } \mu = f(X; W). \quad (28)$$

Here we assume the noise is *homoscedastic* so the variance of the noise $\sigma^2$ is a fixed value, i.e., the noise is independent on the values of features $X$. Thus the parameter is only the mean, $\theta = \{\mu\}$. This assumption may be inappropriate for some applications when the noise is *heteroscedastic*. In such case, we can let the deterministic function $f$ predict both the mean $\mu$ and the variance $\sigma^2$, i.e., $\theta = \{\mu, \sigma^2\}$.

The log-likelihood derived from Equation (28) is

$$\log p(\{x^{(i)}, z^{(i)}\}_{i=1}^n; W) = -\frac{1}{n}\sum_{i=1}^n \frac{\left(z^{(i)} - f(x^{(i)}; W)\right)^2}{2\sigma^2} - \frac{1}{2}\log(2\pi\sigma^2). \quad (29)$$

To maximize the log-likelihood w.r.t. $W$, the loss function can be defined by

$$L(W) = \frac{1}{n}\sum_{i=1}^n \left(z^{(i)} - f(x^{(i)}; W)\right)^2, \quad (30)$$

which is the mean squared error (MSE). Denote the maximum likelihood estimator of $W$ by $\widehat{W}_{ML} = \arg\min_W L(W)$.

Note that the variance $\sigma^2$ is not used in the loss function in Equation (30). To estimate the variance, we can again maximize the likelihood of Equation (29) and obtain the maximum likelihood estimation of $\sigma^2$: $\widehat{\sigma^2}_{ML} = L(\widehat{W}_{ML})$.

## B.2  Gaussian distribution

Gaussian distributions have desired properties, including

(A) *stability*: the family of Gaussian distributions is closed under affine transformations:

$$aZ + b \sim \mathcal{N}(a\mu + b, a^2\sigma^2); \quad (31)$$

(B) *decomposability*: the sum of independent Gaussian random variables is still Gaussian:

$$Z_1 + Z_2 \sim \mathcal{N}(\mu_1 + \mu_2, \sigma_1^2 + \sigma_2^2). \quad (32)$$

Based on these two properties, any linear combination of independent Gaussian random variables is still Gaussian distributed. Thus, several aggregate observations such as the mean observation and the rank observation can be easily modeled and their log-likelihood has an analytical expression, as discussed in Section 5.3 and Section 5.4.

## C Mean observation

In Section 5.3, we discussed a probabilistic model based on the Gaussian distribution for regression via mean observation. In this part, we show that it is possible to use other probabilistic models such as Cauchy distribution and Poisson distribution for regression.

Here, $f_Z$ and $F_Z$ denote the probability density function (PDF) and the cumulative distribution function (CDF) of a real-valued random variable $Z$, respectively.

### C.1 Cauchy distribution

Similarly to the Gaussian distribution, any distribution in the stable distribution family has the property that the linear combination of two independent random variables follows the same distribution up to location and scale parameters. Among the stable distribution family, the Cauchy distribution is studied for the robust regression problem in Liu and Tao [2014], called Cauchy regression.

Concretely, let $Z \sim \mathrm{Cauchy}(a, b) \in \mathbb{R}$ with location parameter $a \in \mathbb{R}$ and scale parameter $b \in (0, \infty)$, where

$$f_Z(z) = \frac{1}{\pi b(1 + z'^2)}, \text{ where } z' = \frac{z - a}{b}, \tag{33}$$

$$F_Z(z) = \frac{1}{\pi} \arctan(z') + \frac{1}{2}. \tag{34}$$

Then the mean $Y = T_{\mathrm{mean}}(Z_{1:K}) = \frac{1}{K} \sum_i^K Z_i$ is still Cauchy distributed:

$$Y \sim \mathrm{Cauchy}(\frac{1}{K} \sum_i^K a_i, \frac{1}{K} \sum_i^K b_i). \tag{35}$$

### C.2 Poisson distribution

In some problems, the true target and the aggregate observations are non-negative integers, e.g., count data. It is different from the normal regression since the support space $\mathcal{Z}$ is discrete.

Poisson distribution is popular for modeling such data, which is used in Law et al. [2018] for spatially aggregated data. Concretely, let $Z \sim \mathrm{Poisson}(\lambda) \in \mathbb{N}_0$ with rate parameter $\lambda \in (0, \infty)$, where

$$f_Z(z) = \frac{\lambda^z e^{-\lambda}}{z!}. \tag{36}$$

Then the sum $Y = T_{\mathrm{sum}}(Z_{1:K}) = \sum_{i=1}^K Z_i$ is still Poisson distributed:

$$Y \sim \mathrm{Poisson}(\sum_{i=1}^K \lambda_i). \tag{37}$$

## D Rank observation

### D.1 Related tasks

Regression via rank observation problem discussed in Section 5.4 is different from learning to rank problem in terms of the type of training data and test data. The exact value of the latent score in learning to rank is not important as long as the relative order is preserved, but the output of a regression function is compared to the ground-truth so that not only the order but also the scale need to be predicted. Another related task is *ordinal regression* [Gutierrez et al., 2015], where the target is usually discrete and ordered, but only the relative order between different values is important. The type of training and test data of several supervised learning tasks is listed in Table 4, and examples of the type of measurements are given in Table 5.

Table 4: **The type of training and test data of supervised learning tasks**

| Task | Training data | Test data |
|---|---|---|
| Classification | categorical | categorical |
| Ordinal Regression | ordinal | ordinal |
| Regression | continuous | continuous |
| Regression via Rank | rank | continuous |
| Learning to Rank | continuous/rank | rank |

Table 5: **Examples of the type of measurements**

| Measurement | Properties | Operators | Example |
|---|---|---|---|
| categorical | discrete, unordered | $=, \neq$ | dog, cat, bear, otter, ... |
| ordinal | discrete, ordered | | excellent, great, good, fair, bad |
| (rank) | (binary) | $>, <$ | (high, low) |
| continuous | continuous | $+, -$ | $-2.71, 0, 3.14, \ldots$ |

## D.2 Gumbel distribution

In Section 5.4, we discussed a probabilistic model based on the Gaussian distribution for regression via rank observation. In this part, we provide other possible probabilistic models where we can also obtain an analytical expression of the log-likelihood.

Gumbel distributed latent scores are used implicitly in RankNet [Burges et al., 2005] and ListNet [Cao et al., 2007]. Concretely, let $Z \sim \text{Gumbel}(\alpha, \beta) \in \mathbb{R}$ with location parameter $\alpha \in \mathbb{R}$ and scale parameter $\beta \in (0, \infty)$. Its PDF and CDF are defined by

$$f_Z(z) = \frac{1}{\beta} e^{-(z' + e^{-z'})}, \text{ where } z' = \frac{z - \alpha}{\beta}, \tag{38}$$

$$F_Z(z) = e^{-e^{-z'}}. \tag{39}$$

Then, if two Gumbel distributed random variable $Z_1$ and $Z_2$ have the same scale $\beta$, we can derive that the difference of two Gumbel distributed random variables follows a logistic distribution:

$$Z_1 - Z_2 \sim \text{Logistic}(m = \alpha_1 - \alpha_2, b = \beta), \tag{40}$$

where

$$f_{Z_1 - Z_2}(d) = \frac{e^{-d'}}{\beta(1 + e^{-d'})^2}, \text{ where } d' = \frac{d - m}{b}, \tag{41}$$

$$F_{Z_1 - Z_2}(d) = \frac{1}{1 + e^{-d'}}. \tag{42}$$

Similarly to Equation (12), we can derive that

$$p(Z_1 > Z_2) = 1 - F_{Z_1 - Z_2}(0) = \frac{1}{1 + e^{-\frac{1}{\beta}(\alpha_1 - \alpha_2)}}. \tag{43}$$

Let $\alpha = s$ be the output of the function $f(X; W)$ and fix the scale $\beta = 1$, we can get the probability of the rank

$$p(Z_1 > Z_2) = \frac{1}{1 + e^{-(s_1 - s_2)}} = \frac{e^{s_1}}{e^{s_1} + e^{s_2}}. \tag{44}$$

The negative log-likelihood is used as the loss function of RankNet [Burges et al., 2005].

Note that the probability is in the form of the logistic function. The scales $\beta$ should be the same for different Gumbel random variables in order to have an analytical expression of the likelihood of the difference.

In some applications, rank observations are in the form of the permutation of a list instead of pairwise comparisons, e.g., $Z_1 > Z_2 > \cdots > Z_K$. A naive way to utilize such data is to decompose it to pairwise comparisons. However, the size of all possible pairs is $\binom{K}{2} = \dfrac{K(K-1)}{2}$, which could be too large if $K$ is large.

On the other hand, for the Gumbel random variables, we could simplify the calculation based on some properties of the Gumbel distribution, including

(A) *max-stability*: the maximum of a set of independent Gumbel distributed random variables is still Gumbel distributed:

Let $\overline{Z} = \max\{Z_1, \ldots, Z_K\}$, then

$$\overline{Z} \sim \mathrm{Gumbel}(\overline{\alpha}, \beta), \text{ where } \overline{\alpha} = \beta \log \sum_{i=1}^{K} \exp \frac{\alpha_i}{\beta}. \tag{45}$$

Therefore,

$$p(Z_1 > \max\{Z_{2:K}\}) = \frac{1}{1 + \sum_{i=2}^{K} e^{-(s_1 - s_i)}} = \frac{e^{s_1}}{\sum_{i=1}^{K} e^{s_i}}; \tag{46}$$

(B) $p(Z_1 > \max\{Z_2, Z_3\}) = p(Z_1 > Z_2 | Z_2 > Z_3) = p(Z_1 > Z_3 | Z_3 > Z_2)$:

*Proof.*

$$p(Z_1 > \max\{Z_2, Z_3\}) = \frac{1}{1 + e^{-\frac{1}{\beta}\left(\alpha_1 - \beta \log\left(e^{\frac{\alpha_2}{\beta}} + e^{\frac{\alpha_3}{\beta}}\right)\right)}} = \frac{e^{\frac{\alpha_1}{\beta}}}{e^{\frac{\alpha_1}{\beta}} + e^{\frac{\alpha_2}{\beta}} + e^{\frac{\alpha_3}{\beta}}}. \tag{47}$$

$$
\begin{aligned}
p(Z_1 > Z_2 > Z_3) &= \int_{-\infty}^{\infty} \int_{-\infty}^{z_1} \int_{-\infty}^{z_2} f_{Z_1}(z_1) f_{Z_2}(z_2) f_{Z_3}(z_3) \, \mathrm{d}z_3 \, \mathrm{d}z_2 \, \mathrm{d}z_1 \\
&= \frac{e^{\frac{\alpha_1}{\beta}}}{e^{\frac{\alpha_1}{\beta}} + e^{\frac{\alpha_2}{\beta}} + e^{\frac{\alpha_3}{\beta}}} \frac{e^{\frac{\alpha_2}{\beta}}}{e^{\frac{\alpha_2}{\beta}} + e^{\frac{\alpha_3}{\beta}}} \\
&= p(Z_1 > \max\{Z_2, Z_3\}) p(Z_2 > Z_3).
\end{aligned} \tag{48}
$$

Therefore $p(Z_1 > Z_2 | Z_2 > Z_3) = \dfrac{p(Z_1 > Z_2 > Z_3)}{p(Z_2 > Z_3)} = p(Z_1 > \max\{Z_2, Z_3\})$.

We can prove the second equation in a similar way.

$\square$

Based on these two properties, it is trivial to generalize it and prove that for Gumbel random variables following property holds:

$$p(Z_1 > Z_2 > \cdots > Z_K) = \prod_{i=1}^{K-1} p(Z_i > \max\{Z_{i+1:K}\}), \tag{49}$$

which is used in the loss function of ListNet [Cao et al., 2007]. However, other distributions such as the Gaussian distribution and the Cauchy distribution do not have these properties, so it is hard to extend pairwise rank observation to listwise rank observation in this way for these distributions.

Note that some special cases of our framework are studied in psychometrics and econometrics, including Bradley-Terry model (pairwise, Gumbel) [Bradley and Terry, 1952], Plackett-Luce model (listwise, Gumbel) [Plackett, 1975], and Thurstone-Mosteller model (pairwise, Gaussian) [Thurstone, 1927], which inspired algorithms such as RankNet [Burges et al., 2005] and ListNet [Cao et al., 2007]. However, the focus is not on the underlying probabilistic model, e.g., the property shown in Equation 46 is treated as an axiom called Luce's choice axiom (LCA) [Luce, 1959] in economics rather than the property of the Gumbel distribution. Our work clarifies the assumptions and the underlying probabilistic model and extends existing work to a variety of aggregate observations.

### D.3 Exponential distribution

In addition to the interpretation of Gumbel distributed latent scores, we provide another probabilistic model based on the exponential distribution that leads to the same learning objective.

Let $Z \sim \text{Exp}(\lambda) \in [0, \infty)$ with inverse scale parameter $\lambda \in (0, \infty)$, where

$$f_Z(z) = \lambda e^{-\lambda z}, \tag{50}$$

$$F_Z(z) = 1 - e^{-\lambda z}. \tag{51}$$

Then, we can derive that the difference of two exponential distributed random variables follows a asymmetric Laplace distribution:

$$Z_1 - Z_2 \sim \text{AsymmetricLaplace}\left(m = 0, \lambda = \sqrt{\lambda_1 \lambda_2}, \kappa = \sqrt{\frac{\lambda_1}{\lambda_2}}\right), \tag{52}$$

where

$$f_{Z_1 - Z_2}(d) = \frac{\lambda_1 \lambda_2}{\lambda_1 + \lambda_2} \begin{cases} e^{\lambda_2 d} & d < 0 \\ e^{-\lambda_1 d} & d \geq 0 \end{cases}, \tag{53}$$

$$F_{Z_1 - Z_2}(d) = \begin{cases} \dfrac{\lambda_1}{\lambda_1 + \lambda_2} e^{\lambda_2 d} & d < 0 \\ 1 - \dfrac{\lambda_2}{\lambda_1 + \lambda_2} e^{-\lambda_1 d} & d \geq 0 \end{cases}. \tag{54}$$

Similarly to Equation 12, we can obtain the probability of the rank

$$p(Z_1 > Z_2) = 1 - F_{Z_1 - Z_2}(0) = \frac{\lambda_2}{\lambda_1 + \lambda_2}. \tag{55}$$

Let $\lambda = e^{-s}$ and let $s$ be the output of the function $f(X; W)$, then

$$p(Z_1 > Z_2) = \frac{1}{1 + e^{-(s_1 - s_2)}} = \frac{e^{s_1}}{e^{s_1} + e^{s_2}}. \tag{56}$$

Note that $-\log(Z) \sim \text{Gumbel}(-\log \lambda, 1)$, which relates to the interpretation of Gumbel distributed latent scores.

If we use other positive function instead of $e^{-s}$ that maps the output of function $f(X; W)$ to the inverse scale parameter $\lambda$, we can derive different learning objectives. The extension is left for future work.

### D.4 Cauchy distribution

As discussed in C.1, Cauchy can be used for robust regression. It also can be used for regression via rank, because the difference of two Cauchy random variables is again Cauchy distributed:

$$Z_1 - Z_2 \sim \text{Cauchy}(a = a_1 - a_2, b = b_1 + b_2), \tag{57}$$

while the probability of the rank is

$$p(Z_1 > Z_2) = 1 - F_{Z_1 - Z_2}(0) = \frac{1}{\pi} \arctan\left(\frac{a_1 - a_2}{b_1 + b_2}\right) + \frac{1}{2}. \tag{58}$$

In conclusion, different distribution assumptions correspond to different sigmoid functions in regression via rank observation and learning to rank problems. Gumbel distributions correspond to the logistic function (the hyperbolic tangent), Gaussian distributions correspond to the error function, and Cauchy distributions correspond to the arctangent function.

For regression via rank observation, the distribution assumption should match the data distribution, especially when we assume that the noise is homoscedastic so the variance is a fixed value. For learning to rank, the Gumbel distribution allows for modeling the maximum of a set of scores explicitly. On the other hand, it is more flexible to model the uncertainty of scores using two-parameter Gaussian distribution or Cauchy distribution if we assume that the noise is heteroscedastic. Because of the fat tails of the Cauchy distribution, it allows for values far from the expected value so that learning could be more robust.

# E   Experiment details

In this section, we provide missing experiment details in Section 6.

## E.1   Classification via triplet comparison (Section 6.1)

**Data.**   We used MNIST [LeCun et al., 1998], Fashion-MNIST [Xiao et al., 2017] and Kuzushiji-MNIST [Clanuwat et al., 2018] datasets without any data augmentation. They all contain $28 \times 28$ grayscale images in 10 classes. The size of the original training dataset is 60000 and the size of the original test dataset is 10000.

**Data preprocessing.**   For both pairwise similarity and triplet comparison data, pairs and triplets of data were sampled randomly with replacement from the original training dataset according to our assumptions in Section 2.2. The size of generated datasets for direct, pairwise, and triplet observations are 60000, 120000, and 180000, respectively.

**Model.**   We used a sequential convolutional neural network as the model $f(X; W)$: Conv2d(#channel = 32), ReLU, Conv2d(#channel = 64), MaxPool2d(size = 2), Dropout(p = 0.25), Linear(#dim = 128), ReLU, Dropout(p = 0.5), Linear(#dim = 10). The kernel size of convolutional layers is 3, and the kernel size of max pooling layer is 2.

**Optimization.**   We used the Adam optimizer with decoupled weight decay regularization [Loshchilov and Hutter, 2019] to train the model. The learning rate is $1 \times 10^{-3}$, the batch size is 128, and the model is trained for 10 epochs.

## E.2   Regression via mean/rank observation on UCI datasets (Section 6.2)

**Data.**   We used 6 benchmark regression datasets from the UCI machine learning repository [Dua and Graff, 2017]. Table 6 shows the statistic of thsese datasets.

Table 6: **Statistic of 6 UCI Benchmark Datasets.**

| Dataset | Dimension | Number of data |
|---|---|---|
| abalone | 10 | 353 |
| airfoil | 5 | 1202 |
| auto-mpg | 7 | 313 |
| concrete | 8 | 824 |
| housing | 13 | 404 |
| power-plant | 4 | 7654 |

**Data preprocessing.**   We split the original datasets into training, validation, and test sets randomly by $60\%$, $20\%$, and $20\%$ for each trial. Only training sets are used for generating the mean and the rank observations. The feature vectors $X$ are standardized to have 0 mean and 1 standard deviation and the targets $Z$ are normalized to have 0 mean using statistics of training sets. For mean observations, the number of sets is the same as the the number of original datasets and each set contains four instances. For rank observations, ten times of data are generated.

**Model.**   We used linear model and gradient boosting machine as the model $f(X; W)$. We used LightGBM [Ke et al., 2017] to implement the gradient boosting machine as the model $f(X; W)$.

**Optimization.**   For the linear model, we used stochastic gradient descent (SGD) with no momentum to train the model. The learning rate is 0.1, the batch size is 256, and the model is trained for 20 epochs. For the gradient boosting machines, the boosting type is gradient boosting decision trees (GBDT), the number of boost rounds is 100, and the number of early stopping rounds is 20.

# F  Additional experiments

In this section, we provide additional experimental results on the datasets different from Section 6. The experiment setup is identical to that of the experiment in Section 6. For the classification task, we used 20 binary classification datasets and 10 multiclass classification datasets. For the regression task, we used additional 20 regression datasets.

## F.1  Classification

We conduct additional experiments on classification via pairwise similarity and triplet comparison in both binary and multiclass classification settings.

### F.1.1  Binary classification

Table 7 shows the statistic of the additional binary classification datasets. The datasets are obtained from OpenML [Vanschoren et al., 2013], the UCI machine learning repository [Dua and Graff, 2017] and LIBSVM [Chang and Lin, 2011]. Table 8 shows the experimental results. For the pairwise similarity case, we can see that contrastive learning and our method can perform comparably to the supervised method in many datasets when we have only two classes. For the triplet comparison, our method clearly outperformed other methods in most cases. Interestingly, Tuplet baseline performs relatively well and arguably better than the Triplet baseline in the binary case. As we will see in the multiclass case, the performance of Tuplet drops miserably compared with other methods.

Table 7: **Statistic of 20 Additional Binary Classification Datasets.**

| Dataset | Dimension | Positive | Negative | Number of data |
|---------|-----------|----------|----------|----------------|
| adult | 104 | 7508 | 22652 | 30160 |
| ayi | 100 | 1385 | 1362 | 2747 |
| banana | 2 | 2376 | 2923 | 5299 |
| codrna | 8 | 162855 | 325709 | 488564 |
| custrev | 100 | 2405 | 1365 | 3770 |
| ijcnn | 22 | 18418 | 173262 | 191680 |
| image | 18 | 1187 | 898 | 2085 |
| magic | 10 | 12330 | 6687 | 19017 |
| mpqa | 100 | 3311 | 7291 | 10602 |
| mushroom | 98 | 3487 | 2155 | 5642 |
| phishing | 30 | 6157 | 4896 | 11053 |
| phoneme | 5 | 3817 | 1586 | 5403 |
| ringnorm | 20 | 3663 | 3736 | 7399 |
| rt-polarity | 100 | 5331 | 5330 | 10661 |
| spambase | 57 | 1811 | 2788 | 4599 |
| splice | 60 | 1344 | 1646 | 2990 |
| subj | 100 | 5000 | 4999 | 9999 |
| susy | 18 | 45974 | 54024 | 99998 |
| twonorm | 20 | 3703 | 3696 | 7399 |
| w8a | 300 | 1933 | 62766 | 64699 |

Table 8: **Classification via pairwise similarity and triplet comparison.** Means and standard deviations of accuracy in percentage for 10 trials are reported.

| Dataset | Unsupervised | Pairwise Similarity | | | Triplet Comparison | | | Supervised |
|---|---|---|---|---|---|---|---|---|
| | | Siamese | Contrastive | Ours | Tuplet | Triplet | Ours | |
| adult | 71.38 | 69.91 | 81.86 | **84.15** | 73.73 | 57.89 | **75.49** | 85.08 |
| | (0.29) | (11.11) | (3.57) | (**0.43**) | (0.59) | (7.07) | (**0.63**) | (0.29) |
| ayi | 53.25 | 66.93 | **76.67** | **77.65** | 52.16 | **54.22** | **54.51** | 78.47 |
| | (1.58) | (10.90) | (**1.20**) | (**1.68**) | (2.09) | (**1.40**) | (**3.28**) | (1.40) |
| banana | 56.16 | 70.22 | **89.80** | 89.47 | 53.26 | **56.75** | **57.53** | 88.42 |
| | (0.94) | (16.99) | (**1.20**) | (**0.79**) | (2.50) | (**6.25**) | (**5.36**) | (1.11) |
| codrna | 55.58 | 59.23 | 96.38 | **96.57** | 61.06 | 59.28 | **66.72** | 96.47 |
| | (0.10) | (12.33) | (0.08) | (**0.11**) | (2.43) | (8.68) | (**0.18**) | (0.07) |
| custrev | 51.70 | 60.76 | 70.84 | **74.56** | 53.24 | 52.33 | **60.92** | 75.32 |
| | (1.26) | (9.07) | (1.27) | (**1.10**) | (1.67) | (2.13) | (**4.67**) | (2.13) |
| ijcnn | 58.39 | 58.48 | **99.13** | 98.98 | 74.02 | 52.88 | **92.15** | 99.08 |
| | (1.65) | (13.02) | (**0.04**) | (0.19) | (10.74) | (4.34) | (**0.56**) | (0.05) |
| image | 60.94 | 75.44 | **94.10** | 93.81 | 54.51 | **56.09** | **59.02** | 89.76 |
| | (1.44) | (18.45) | (**0.84**) | (**1.16**) | (3.15) | (**3.31**) | (**7.21**) | (0.99) |
| magic | 54.55 | 77.18 | 86.44 | **86.93** | **67.49** | 58.79 | 65.94 | 86.31 |
| | (0.81) | (12.22) | (0.59) | (**0.52**) | (**2.50**) | (5.27) | (1.08) | (0.41) |
| mpqa | 62.56 | 59.38 | **84.76** | 84.60 | 53.33 | 53.18 | **73.06** | 85.89 |
| | (0.86) | (13.56) | (**0.68**) | (**0.87**) | (2.73) | (1.24) | (**1.41**) | (0.67) |
| mushroom | 85.14 | 82.17 | **99.98** | 99.96 | 61.37 | 56.47 | **88.46** | 99.97 |
| | (0.75) | (21.95) | (**0.04**) | (**0.08**) | (4.28) | (6.04) | (**8.43**) | (0.08) |
| phishing | 54.80 | 77.65 | **96.07** | 96.43 | 55.64 | 57.08 | **73.28** | 95.25 |
| | (0.82) | (21.31) | (**0.37**) | (**0.56**) | (3.02) | (4.82) | (**2.44**) | (0.35) |
| phoneme | 68.24 | 60.64 | 80.82 | **83.59** | 59.16 | 62.16 | **71.39** | 81.06 |
| | (0.51) | (9.46) | (1.00) | (**1.27**) | (8.19) | (8.20) | (**0.72**) | (0.75) |
| ringnorm | 76.05 | 82.95 | 97.20 | **98.07** | **72.54** | 58.82 | **67.92** | 97.97 |
| | (1.05) | (20.56) | (0.39) | (**0.22**) | (**4.11**) | (8.26) | (**9.70**) | (0.31) |
| rt-polarity | 51.99 | 57.44 | 69.73 | **71.04** | 52.26 | **52.82** | **54.20** | 72.37 |
| | (0.82) | (8.92) | (0.71) | (**0.53**) | (1.64) | (**2.27**) | (**2.53**) | (1.01) |
| spambase | 60.04 | 81.70 | 93.61 | **94.23** | 57.66 | 60.00 | **71.85** | 93.73 |
| | (1.17) | (18.05) | (0.44) | (**0.48**) | (1.50) | (6.13) | (**7.39**) | (0.88) |
| splice | 66.69 | 70.15 | 87.79 | **90.95** | 53.16 | 54.23 | **59.05** | 89.48 |
| | (1.25) | (19.07) | (1.16) | (**1.23**) | (1.59) | (2.78) | (**7.63**) | (1.11) |
| subj | 82.28 | 74.99 | 87.48 | **88.79** | 53.12 | 54.79 | **64.36** | 89.18 |
| | (0.86) | (16.71) | (0.56) | (**0.53**) | (2.42) | (3.60) | (**6.80**) | (0.85) |
| susy | 67.19 | 66.07 | 79.48 | **79.78** | **58.68** | 53.67 | 54.65 | 79.87 |
| | (0.27) | (12.93) | (0.37) | (**0.20**) | (**2.84**) | (2.93) | (4.60) | (0.26) |
| twonorm | 97.66 | 84.36 | **97.64** | **97.64** | 60.60 | 58.16 | **63.47** | 97.65 |
| | (0.36) | (20.17) | (**0.37**) | (**0.42**) | (**8.76**) | (2.99) | (**9.13**) | (0.37) |
| w8a | 96.38 | 61.49 | **98.00** | 98.26 | 96.97 | 57.98 | **97.78** | 99.05 |
| | (1.79) | (12.33) | (**0.59**) | (**1.00**) | (0.15) | (2.36) | (**0.15**) | (0.04) |

### F.1.2 Multiclass classification

Table 9 shows the statistic of the additional multiclass classification datasets. The datasets are obtained from OpenML [Vanschoren et al., 2013]. In Table 10, our methods nicely outperformed representation learning based methods in all cases for triplet comparison. For pairwise similarity, contrastive learning outperformed our method in cardiotocography and isolet datasets, while our method clearly outperformed other methods on artificial-character, covertypy, gas-drift, and satimage.

Table 9: **Statistic of 10 Additional Multiclass Datasets.**

| Dataset | Dimension | Number of classes | Number of data |
|---|---|---|---|
| artificial-character | 7 | 10 | 10217 |
| cardiotocography | 35 | 10 | 2125 |
| covertype | 54 | 7 | 581011 |
| gas-drift | 128 | 6 | 13909 |
| isolet | 617 | 26 | 7796 |
| japanesevowels | 14 | 9 | 9960 |
| letter | 16 | 26 | 19999 |
| pendigits | 16 | 10 | 10991 |
| satimage | 36 | 6 | 6429 |
| vehicle | 18 | 4 | 845 |

Table 10: **Classification via pairwise similarity and triplet comparison.** Means and standard deviations of accuracy in percentage for 10 trials are reported.

| Dataset | Unsupervised | Pairwise Similarity | | | Triplet Comparison | | | Supervised |
|---|---|---|---|---|---|---|---|---|
| | | Siamese | Contrastive | Ours | Tuplet | Triplet | Ours | |
| artificial-character | 22.05 | 34.93 | 46.31 | **50.29** | 22.18 | 23.04 | **47.71** | 57.71 |
| | (0.75) | (2.76) | (0.78) | (**1.19**) | (1.97) | (1.90) | (**1.56**) | (1.00) |
| cardiotocography | 88.14 | 85.88 | **99.88** | 95.39 | 36.89 | 37.27 | **94.35** | 99.95 |
| | (6.57) | (15.75) | (**0.24**) | (1.73) | (2.90) | (4.52) | (**6.76**) | (0.09) |
| covertype | 36.24 | 47.81 | 54.17 | **81.15** | 36.14 | 25.17 | **64.85** | 84.69 |
| | (3.97) | (6.17) | (4.03) | (**1.09**) | (3.80) | (2.89) | (**7.37**) | (0.35) |
| gas-drift | 37.29 | 61.32 | 90.40 | **98.40** | 26.32 | 38.71 | **82.60** | 98.56 |
| | (0.78) | (15.06) | (7.57) | (**0.59**) | (1.58) | (4.86) | (**5.55**) | (0.33) |
| isolet | 55.69 | 73.04 | **86.24** | 78.79 | 17.32 | 22.38 | **57.73** | 96.12 |
| | (2.38) | (1.88) | (**2.27**) | (3.44) | (1.02) | (2.90) | (**3.14**) | (0.57) |
| japanesevowels | 38.42 | 73.29 | **96.25** | **96.61** | 27.80 | 31.69 | **94.66** | 96.55 |
| | (1.00) | (12.45) | (**0.37**) | (**0.55**) | (2.68) | (3.49) | (**0.75**) | (0.38) |
| letter | 29.06 | 48.43 | **73.59** | **73.86** | 21.08 | 23.12 | **64.56** | 89.41 |
| | (1.74) | (5.10) | (**1.65**) | (**2.14**) | (1.46) | (1.95) | (**1.57**) | (0.57) |
| pendigits | 68.22 | 83.17 | **99.03** | **99.02** | 43.01 | 41.01 | **97.24** | 98.68 |
| | (2.74) | (8.09) | (**0.18**) | (**0.22**) | (3.68) | (3.08) | (**2.12**) | (0.20) |
| satimage | 64.32 | 65.41 | 83.52 | **88.23** | 44.28 | 47.30 | **84.33** | 87.82 |
| | (6.01) | (4.21) | (1.87) | (**0.52**) | (4.03) | (6.61) | (**0.84**) | (0.84) |
| vehicle | 38.40 | 46.21 | **69.35** | 68.34 | 35.92 | 41.60 | **57.10** | 69.23 |
| | (1.70) | (7.79) | (**5.35**) | (**4.90**) | (3.39) | (5.63) | (**7.28**) | (2.66) |

### F.2 Regression

#### F.2.1 Varying size of sets in regression via mean observation

Table 11 shows the experimental result when the number of sample points that are aggregated in each set (i.e., $K$) is changed, where $K \in \{2, 4, 8, 16\}$. The result shows that larger sets lead to slightly worse performance, which illustrates the trade-off between the quality of labels and the costs/privacy conservation. However, the performance difference is still marginal.

Table 11: **Varying $K$ in regression via mean observation on UCI benchmark datasets.** Means and standard deviations of MSE for 10 trials are reported. We compare linear regression (LR) and gradient boosting machines (GBM) as the regression function.

| Dataset | LR | | | | GBM | | | |
|---|---|---|---|---|---|---|---|---|
| | 2 | 4 | 8 | 16 | 2 | 4 | 8 | 16 |
| abalone | 5.52 | 5.33 | 5.60 | 5.89 | 4.68 | 4.89 | 4.70 | 4.98 |
| | (0.4) | (0.4) | (0.5) | (0.4) | (0.3) | (0.4) | (0.2) | (0.3) |
| airfoil | 23.08 | 24.00 | 25.78 | 30.91 | 4.87 | 4.97 | 4.98 | 4.98 |
| | (2.0) | (2.2) | (1.5) | (1.7) | (0.5) | (0.7) | (0.7) | (0.8) |
| auto-mpg | 13.54 | 12.67 | 15.00 | 21.52 | 10.03 | 9.88 | 10.46 | 10.68 |
| | (2.1) | (2.3) | (3.3) | (4.6) | (2.6) | (1.6) | (2.5) | (2.0) |
| concrete | 120.36 | 121.74 | 119.78 | 149.48 | 31.98 | 32.01 | 32.55 | 31.71 |
| | (12.9) | (13.3) | (8.9) | (12.8) | (3.8) | (3.7) | (3.4) | (4.4) |
| housing | 27.82 | 28.36 | 20.90 | 34.67 | 13.27 | 17.97 | 16.63 | 15.10 |
| | (9.5) | (8.1) | (4.4) | (6.8) | (2.8) | (4.4) | (4.4) | (5.1) |
| power-plant | 21.02 | 21.13 | 21.61 | 24.12 | 13.17 | 13.04 | 14.03 | 13.49 |
| | (1.0) | (1.4) | (1.0) | (0.8) | (0.7) | (0.7) | (1.2) | (0.9) |

#### F.2.2 Mean/rank observation

Table 12 shows the statistic of the additional regression datasets. The datasets are obtained from OpenML [Vanschoren et al., 2013] and the UCI machine learning repository [Dua and Graff, 2017]. Table 13 shows the mean squared error results for each method. For the regression via mean observation, our method based on linear regression outperformed the linear regression baseline consistently in all cases, similarly for the gradient boosting machine case. Sometimes they are comparable to the supervised method. For the regression via rank observation, it is interesting to see that our method based on a Gaussian distribution failed in house-16h and house-8l datasets. We hypothesize that it is because the target distribution is heavy-tailed in these datasets.

Table 12: **Statistic of 20 Additional Regression Datasets.**

| Dataset | Dimension | Min | Max | Number of data |
|---|---|---|---|---|
| 2d-planes | 10 | -12.6943 | 12.2026 | 40767 |
| bank-32nh | 32 | 0 | 82 | 8191 |
| bank-8fm | 8 | 0 | 80.2263 | 8191 |
| bike-sharing | 33 | 0.022 | 8.71 | 730 |
| cpu-act | 21 | 0 | 99 | 8191 |
| diabetes | 10 | 25 | 346 | 441 |
| elevator | 6 | -15 | 15.1 | 9516 |
| fried | 10 | -1.228 | 30.522 | 40767 |
| house-16h | 16 | 0 | 50 | 22783 |
| house-8l | 8 | 0 | 50 | 22783 |
| insurance-charge | 11 | 1.12 | 63.8 | 1336 |
| kin-8nm | 8 | 4.016538 | 145.8521 | 8191 |
| puma-8nh | 8 | -12.4153 | 11.87619 | 8191 |
| real-estate | 6 | 7.6 | 117.5 | 413 |
| rmftsa-ladata | 10 | 4.15 | 30.43 | 507 |
| space-ga100 | 6 | -305.704 | 10.00835 | 3106 |
| stock | 9 | 34 | 62 | 949 |
| wine-quality | 11 | 3 | 9 | 6496 |
| wine-red | 11 | 3 | 8 | 1598 |
| wine-white | 11 | 3 | 9 | 4897 |

Table 13: **Regression via mean observation and rank observation.** Means and standard deviations of error variance (rank observations) or MSE (otherwise) for 10 trials are reported.

| Dataset | Mean Observation | | | | Rank Observation | | | | Supervised | |
|---|---|---|---|---|---|---|---|---|---|---|
| | Baseline | | Ours | | RankNet, Gumbel | | Ours, Gaussian | | | |
| | LR | GBM | LR | GBM | LR | GBM | LR | GBM | LR | GBM |
| 2d-planes | 13.49 (0.20) | 11.40 (0.20) | 5.70 (0.00) | **1.01** (**0.00**) | 8.86 (0.10) | 19.14 (0.00) | 5.72 (0.10) | **1.03** (**0.00**) | 5.72 (0.1) | 1.00 (0.0) |
| bank-32nh | 112.42 (5.70) | 111.55 (4.00) | **69.30** (**2.60**) | 70.32 (**3.70**) | 123.63 (6.50) | 142.97 (6.40) | **71.25** (**3.90**) | 87.79 (4.10) | 70.62 (3.1) | 68.83 (4.2) |
| bank-8fm | 134.99 (5.70) | 133.56 (4.50) | 15.23 (0.90) | **10.07** (**0.30**) | 86.92 (2.60) | 230.19 (6.40) | 34.19 (**1.20**) | 33.17 (**1.50**) | 15.45 (0.7) | 9.21 (0.5) |
| bike-sharing | 2.36 (0.20) | 2.40 (0.20) | 0.77 (0.10) | **0.63** (**0.10**) | 1.61 (0.30) | 3.79 (0.30) | 0.68 (0.10) | **0.57** (**0.10**) | 0.64 (0.1) | 0.45 (0.1) |
| cpu-act | 235.22 (23.70) | 200.64 (18.40) | 98.20 (10.20) | **6.10** (**0.40**) | 231.24 (10.60) | 343.39 (33.40) | 26484.33 (18656.30) | **145.88** (**12.70**) | 102.68 (17.1) | 5.07 (0.2) |
| diabetes | 4688.70 (541.30) | 4937.20 (687.70) | **3335.22** (**430.40**) | 3746.14 (**782.30**) | 5765.28 (303.80) | 5918.11 (700.90) | 5968.57 (501.70) | **3771.73** (**440.60**) | 3323.98 (491.2) | 3462.89 (464.6) |
| elevator | 18.91 (0.80) | 18.21 (0.50) | 13.03 (0.40) | **10.87** (**0.30**) | 17.38 (0.40) | 23.99 (0.80) | 12.93 (0.30) | **10.65** (**0.30**) | 12.83 (0.3) | 10.57 (0.3) |
| fried | 17.15 (0.20) | 14.84 (0.20) | 6.92 (0.10) | **1.28** (**0.00**) | 12.04 (0.20) | 25.03 (0.30) | 6.94 (0.10) | **1.62** (**0.00**) | 7.03 (0.1) | 1.21 (0.0) |
| house-16h | 24.33 (2.20) | 20.21 (1.20) | 21.28 (1.20) | **9.69** (**1.10**) | 22.69 (1.50) | 27.26 (1.80) | 1462200.27 (963977.00) | **17.52** (**1.50**) | 22.33 (1.9) | 9.88 (0.8) |
| house-8l | 23.71 (1.50) | 20.08 (1.10) | 17.95 (1.40) | **8.71** (**0.40**) | 21.16 (1.20) | 28.51 (1.30) | 197143.11 (199581.40) | **15.99** (**1.40**) | 18.04 (1.3) | 8.80 (0.6) |
| insurance-charge | 95.74 (8.80) | 91.59 (8.40) | 36.46 (4.50) | **25.26** (**3.10**) | 99.42 (10.00) | 143.41 (13.50) | 68.98 (7.40) | **46.36** (**7.50**) | 40.66 (6.2) | 20.72 (3.3) |
| kin-8nm | 566.46 (14.40) | 489.96 (11.40) | 406.42 (14.10) | **192.96** (**8.00**) | 660.73 (18.70) | 689.58 (19.60) | 523.07 (9.30) | **187.33** (**10.00**) | 411.87 (12.4) | 169.30 (7.4) |
| puma-8nh | 26.68 (0.60) | 22.64 (0.40) | 19.60 (0.50) | **10.77** (**0.60**) | 26.31 (0.40) | 31.56 (0.40) | 19.82 (0.90) | **10.36** (**0.40**) | 19.90 (0.6) | 10.40 (0.4) |
| real-estate | 155.19 (42.10) | 118.97 (34.70) | **68.82** (**25.10**) | 64.47 (**27.60**) | 130.77 (28.30) | 194.92 (51.70) | 128.28 (27.90) | **61.10** (**21.70**) | 92.49 (37.3) | 54.94 (26.0) |
| rmftsa-ladata | 6.93 (1.00) | 5.19 (1.00) | **3.77** (**1.00**) | 4.34 (**0.90**) | 5.59 (2.10) | 8.94 (3.50) | **3.89** (**1.60**) | 5.56 (1.20) | 3.99 (1.1) | 4.00 (1.1) |
| space-ga100 | 313.80 (40.60) | 301.18 (49.40) | 237.95 (45.60) | **127.45** (**12.50**) | 366.00 (31.30) | 384.73 (54.70) | 300.72 (52.80) | **162.05** (**34.20**) | 221.42 (47.8) | 115.54 (18.7) |
| stock | 26.02 (1.90) | 24.76 (1.30) | 6.50 (0.50) | **1.44** (**0.30**) | 17.35 (1.10) | 43.21 (2.40) | 13.19 (1.00) | **1.32** (**0.20**) | 5.86 (0.7) | 0.79 (0.1) |
| wine-quality | 0.66 (0.00) | 0.62 (0.00) | 0.55 (0.00) | **0.45** (**0.00**) | 0.63 (0.00) | 0.77 (0.00) | 0.55 (0.00) | **0.44** (**0.00**) | 0.55 (0.0) | 0.43 (0.0) |
| wine-red | 0.55 (0.00) | 0.53 (0.00) | **0.40** (**0.00**) | 0.40 (**0.00**) | 0.54 (0.10) | 0.64 (0.00) | 0.44 (0.00) | **0.37** (**0.00**) | 0.41 (0.0) | 0.36 (0.0) |
| wine-white | 0.69 (0.00) | 0.63 (0.00) | 0.56 (0.00) | **0.46** (**0.00**) | 0.61 (0.00) | 0.81 (0.00) | 0.57 (0.00) | **0.45** (**0.00**) | 0.58 (0.0) | 0.44 (0.0) |