[Reviews · NeurIPS 2020]

Review 1

Summary and Contributions: - The paper considers the problem of learning from aggregate observations, where the individual labels are not available, but an aggregate of these are available through some aggregation function (known). - The paper proposes a general setting for dealing with different aggregation, and provide multiple applications - pairwise similarity/triplet comparison (classification) and rank observation (regression).

Strengths: - While their probabilistic framework is not quite new, previously explored in a similar fashion in [1] (ignoring the GP base model), the precise setup is novel, as it covers a wider setting, and the authors propose a wide range of applications ([1] mostly do so for exponential family). - In particular, the idea of consistency of learning from aggregate observations is interesting, and the theory provided is useful for the precise formulation of the problem (for maximum likelihood at least). - The authors proposal of the 3 applications is mostly new (though maybe not the regression via mean, as this was explored also in [1]), and they provide their MLE analysis for these applications. - For these applications, the authors provide a large set of experimental results to demonstrate their method is superior vs their baselines. [1] Variational learning on aggregate outputs with Gaussian processesI t HC Law, D Sejdinovic, E Cameron, T Lucas, S Flaxman, K Battle, K Fukumizu Advances in Neural Information Processing Systems, 2018, 6081-6091

Weaknesses: If the framework is available for classification (LLP - learning from label proportions), I would like to see some experimental results on that, so that we can see how the method compares against stronger baselines. The baselines currently are quite weak in the experiments, however I note that there might not be too much related work on these applications in this setting.

Correctness: I did not check the proofs very detailedly, but however the authors proposal make sense and the empirical methodology is fine.

Clarity: I think the paper is well written in general.

Relation to Prior Work: I am not too familiar on the baselines on the applications that the authors proposed, however from related work it seems to be mostly clear.

Reproducibility: Yes

Additional Feedback: Is there a reason the authors use linear regression as one of the methodology for the experiments? I presume any differential model can be used, since we are just using MLE. ------- I have updated my score to a 7 after rebuttal and reading other reviewer comments. The overall framework is useful, and there seems to be applications previously not thought about.


Review 2

Summary and Contributions: The authors formalize a consistent estimation procedure for learning from aggregate observations. This is an important and recently overlooked problem. The authors propose a classical MLE for this. Evaluations make sense. --- My original rating still stands.

Strengths: I like this paper. This problem of learning from pool/bag of observations where one cannot identify each instance separately (thereby having some information loss) is very important in weakly labeled learning; self supervised learning; personalization etc. The presentation of the model and evaluations are good.

Weaknesses: Presentation can be improved a bit. And the evaluations may be expanded, but nevertheless this is already a good paper. Passed the threshold of the conference.

Correctness: Yes.

Clarity: Yes. Very well written.

Relation to Prior Work: Yes.

Reproducibility: Yes

Additional Feedback: As stated about this is an important problem; learning from bags of data points with a cumulative descriptive statistic representing them. And this is clearly a longer journal compressed to a 8 page format. To that end, the current presentation can be improved. Firstly adding few remarks to the main technical result is useful. Second, how does this setup differ from the weakly labeled setting (or does the proposed estimation provide some guarantees for such weakly labeled aggregation based learners).


Review 3

Summary and Contributions: Substantially expands the previous "aggregate function learning" settings meant for binary classification to multiclass and regression.

Strengths: Formulation is clean, very well presented, and covers several aggregation scenarios.

Weaknesses: It would be nice to present some real life problems where the settings they model naturally arise. I can think of one situation but there Assumption 2 does not hold. Currently they reformulate som multi-class problems to fit their scenarios. So the exercise seems academic without immediate industry impact.

Correctness: yes and yes

Clarity: very well

Relation to Prior Work: yes

Reproducibility: Yes

Additional Feedback: UPDATE: I have read and taken into account the rebuttal, as well as any ensuing discussions.


Review 4

Summary and Contributions: This paper presents a general framework for learning classification/regression models from aggregate observations, e.g., the similarity, the mean, and the rank. The model parameters can be estimated via the maximum likelihood principle. The characteristics of the solution are analyzed theoretically.

Strengths: The problem that the authors are working on is important and fundamental. The formulation is general, and I think it is the first attempt to consider theoretical aspects for learning from aggregate observations.

Weaknesses: Some problems discussed in this work have been addressed previously, and this work is its generalization. It is important and useful work, but the task, i.e., learning from aggregate observations, might not be novel.

Correctness: I think this paper is technically sound, and the experimental section is well-organized.

Clarity: The paper is well presented, easily understandable.

Relation to Prior Work: Important related works are well presented. Meanwhile, there are other studies that are close to the motivation for this work. If necessary, it might be helpful to add a discussion of the relevance of these studies, called *ecological inference*. ・Flaxman, S. R.; Wang, Y. X.; and Smola, A. J. 2015. Who supported Obama in 2012?: Ecological inference through distribution regression. In KDD, 289–298. ・D. R. Sheldon and T. G. Dietterich. Collective graphical models. In NIPS, pages 1161–1169, 2011.

Reproducibility: Yes

Additional Feedback: Please respond to the above comments. Additional questions. 1. The experiments showed a significant improvement in accuracy in the triplet comparison, but I did not understand why. Also, the significant improvement in the triplet comparison is a good result, but what are some specific/practical examples where the triplet comparison is given? 2. In the regression task, how did you select multiple samples that are aggregated? randomly selected? After author feedback: I appreciate the authors' feedback. The task addressed in this work is not novel, but I think it has good contributions, that is, 1) provide a clear and general formulation for learning from aggregate observations, 2) discuss the theoretical aspects of the MLE in Section 4. For publication, I would like the authors to discuss the following issues: 1) As the other reviewer mentioned, Assumption 2 seems restrictive in some situations. In the regression task, the authors said in the author response that the samples aggregated were picked randomly. However, for example, spatial data could be often aggregated over appropriate regions, so that the samples in each region are correlated. The authors mentioned its limitation in Line 86-87 in the manuscript, but it would be better to emphasize this limitation from theoretical and practical aspects. 2) I think the authors cited related works sufficiently in the field of machine learning, but related works actually exist in other fields such as statistics/geo-statistics. One of them is the concept, ecological fallacy or ecological inference (mentioned in my review), which addresses the problem of learning individual-level model from aggregate observations. In this line, ML-specific tasks (e.g., classification) have not been addressed, but it is related to this work. So, I recommend the authors will include this related concept in related works.

[Author Response · NeurIPS 2020]

We thank the reviewers for helpful comments and suggestions. We will address the concerns raised by the reviewers.

(R1:Q1) On using our framework for learning from label proportions (LLP).

(R1:A1) Our proposed framework is applicable for tackling learning from label proportions, even for the multiclass
case, by using class proportions as aggregated labels. However, due to space limitation and the fact that LLP has been
explored extensively, we would like to focus on other problem settings to expand the usage of learning from aggregate
observations and provide the theoretical foundation of a more general case. Nevertheless, we agree that it is interesting
to see the performance of this framework compared with other LLP methods to see the competency of our framework.
We will add more explanations of LLP, potentially in Appendix due to lack of space.

(R1:Q2) Baselines are quite weak in the experiments, however I note that there might not be too much related work.

(R1:A2) As pointed out, related work that can be used as baselines for our experiments are quite limited. For example,
we are not aware of any methods for multiclass classification from triplet comparison data. We tried to come up
with several baselines and found that a representation learning method is reasonable and its performance is quite
reasonable. It worked quite well in the pairwise comparison case but failed to work well in the triplet case which
might be because more data are needed. For regression via mean observation, [1] is the most related as suggested. The
difference is that [1] used Gaussian processes and variational inference. We believe both frameworks have different
advantages/disadvantages such as the variety of model choices, scalability, or the uncertainty measure. We will add
such discussion in the final version.

(R1:Q3) Why linear regression is used as one of the methods in the experiments?

(R1:A3) As R1 suggested, we can use any differentiable model. Linear regression was used because it is one of the
standard models for regression on these datasets. Moreover, it is insightful to see the difference in performance between
a linear model and a more complex model with the same objective function. Thus, we implemented the proposed
objective and the baseline objective on both the linear model and a gradient boosting machine (GBM). We will add
more discussion on the choice of models.

(R2:Q4) How does this setup differ from the weakly labeled setting?

(R2:A4) Our problem setting can be regarded as a weakly-supervised learning problem, where only a group-level label
is observed although we want to predict a label for an individual instance. It is different from many weakly-labeled
settings in the literature (e.g., partial labels, complementary labels, positive-unlabeled learning, noisy labels) in the
sense that individual labels are given in those settings although they are weak (i.e., not clean fully-supervised).

(R2:Q5) On how to improve paper's presentation

(R2:A5) Thank you. We will provide explanations to give key ideas how to interpret our results and why they are useful.

(R3:Q6) On the practicality of Assumption 2

(R3:A6) We admit that it is possible that Assumption 2 is violated in real-world problems. Thus, it is interesting to relax
Assumption 2 and investigate the situation when this assumption does not hold. For example, we may try to explore a
new framework that relies on another assumption that is more practical in some settings. Then, a practitioner can select
an appropriate method depending on their problem of interests. We believe there are many issues to be discussed when
going beyond this assumption and it is a good future direction. we will discuss these issues as a future work in the final
version.

(R4:Q7) Why the proposed method is much better in classification from triplet comparisons?

(R4:A7) One explanation is we might need much more data to learn a reasonable representation compared with simply
learning a probabilistic classifier to separate between classes. In Appendix E, we also showed the performance in the
binary classification task and found that the baseline can be quite competitive for some datasets. But in the multiclass
cases especially when the number of classes is quite high, baselines become much weaker than the proposed method.
We will add more discussion in the final version.

(R4:Q8) What are some specific/practical examples where the triplet comparison is given?

(R4:A8) Examples include the sensor network problem and search engine query logs, which were discussed in [12].
Triplets has been used a lot for representation learning but not classification (maybe due to lack of methods). We will
include this issue in Introduction.

(R4:Q9) How did you select multiple samples that are aggregated in the experiments?

(R4:A9) It was randomly selected. We are aware that this way may not be ideal. To make up for that, we did experiments
on many datasets (59 datasets including Appendix E). We will add more explanations in the final version.

[Meta-Review · NeurIPS 2020]

There is a consensus among the knowledgeable reviewers that this is a good paper that addresses one of the important problems. No major concerns was raised during the discussion and the authors provided a rebuttal that helped clarify some of the reviewers' questions. The paper is therefore accepted as poster. Nevertheless, one of the concerns that came up during the discussion is that Assumption 2 seems restrictive in some situations. The authors discussed this limitation in Line 86-87 in the manuscript, but it would improve the paper significantly if the authors could emphasize this limitation from theoretical and practical aspects in their camera-ready version. The reviewers also gave suggestions on how to improve the paper in their reviews. Please also take them into consideration when preparing the camera-ready version.